# Estrogen Receptors and Estrogen-Induced Uterine Vasodilation in Pregnancy

**DOI:** 10.3390/ijms21124349

**Published:** 2020-06-18

**Authors:** Jin Bai, Qian-Rong Qi, Yan Li, Robert Day, Josh Makhoul, Ronald R. Magness, Dong-bao Chen

**Affiliations:** 1Department of Obstetrics & Gynecology, University of California, Irvine, CA 92697, USA; baij3@hs.uci.edu (J.B.); qianronq@hs.uci.edu (Q.-R.Q.); liy60@hs.uci.edu (Y.L.); dayr1@uci.edu (R.D.); joshua.makhoul@gmail.com (J.M.); 2Department of Obstetrics & Gynecology, University of South Florida, Tampa, FL 33612, USA; rmagness@usf.edu

**Keywords:** estrogen receptors, estrogens, nitric oxide, hydrogen sulfide, vasodilatation, uterine artery, pregnancy, preeclampsia

## Abstract

Normal pregnancy is associated with dramatic increases in uterine blood flow to facilitate the bidirectional maternal–fetal exchanges of respiratory gases and to provide sole nutrient support for fetal growth and survival. The mechanism(s) underlying pregnancy-associated uterine vasodilation remain incompletely understood, but this is associated with elevated estrogens, which stimulate specific estrogen receptor (ER)-dependent vasodilator production in the uterine artery (UA). The classical ERs (ERα and ERβ) and the plasma-bound G protein-coupled ER (GPR30/GPER) are expressed in UA endothelial cells and smooth muscle cells, mediating the vasodilatory effects of estrogens through genomic and/or nongenomic pathways that are likely epigenetically modified. The activation of these three ERs by estrogens enhances the endothelial production of nitric oxide (NO), which has been shown to play a key role in uterine vasodilation during pregnancy. However, the local blockade of NO biosynthesis only partially attenuates estrogen-induced and pregnancy-associated uterine vasodilation, suggesting that mechanisms other than NO exist to mediate uterine vasodilation. In this review, we summarize the literature on the role of NO in ER-mediated mechanisms controlling estrogen-induced and pregnancy-associated uterine vasodilation and our recent work on a “new” UA vasodilator hydrogen sulfide (H_2_S) that has dramatically changed our view of how estrogens regulate uterine vasodilation in pregnancy.

## 1. Introduction

During the ovarian cycle, follicular development accompanies phasic persistent hyperemia in the uterine vascular bed. Dramatic hyperemia occurs just prior to and during estrus when preovulatory follicles produce the largest quantity of estrogens [1]. Uterine blood flow (UtBF) sharply elevates during the same period, returns to basal levels after estrus, and remains low during the luteal phase of the ovarian cycle [2] when estrogen levels are at their nadir and progesterone is elevated [3]. During normal pregnancy, maternal hemodynamics change dramatically, characterized by mild decreased systemic blood pressure and a larger fall in systemic vascular resistance, associated with increased heart rate, stroke volume, cardiac output, and blood volume [4,5,6,7,8,9,10,11]. The largest portion (~15–20%) of the increased cardiac output is redistributed to the aggressively developing uteroplacental vascular bed, which is reflected by substantial pregnancy-associated periodical rises in UtBF [8,12,13,14]. Specifically, two peak rises in UtBF that occur during the second and third trimesters correlate, respectively, to the completion of placentation and rapid fetal growth [13,14]. Noteworthily, circulating and/or local estrogen levels are, coincidentally, elevated with these two UtBF peaks [15,16,17], suggesting that estrogens may play a pivotal role in regulating UtBF during normal pregnancy.

Markee (1932) was the first to demonstrate the vascular effects of estrogens; he showed that treatment with crude follicular estrogen extracts results in the vasodilatation (hyperemia) of uterine endometrial tissue transplanted to the anterior chamber of the monkey eye [18]. There are now data convincingly showing that estrogens are potent vasodilators that cause blood flow to rise in several organs throughout the body with the greatest effects occurring in reproductive tissues, especially the uterus [9,19,20,21]. In ovariectomized (OVX) nonpregnant ewes, daily estradiol-17β (E_2_β) increases basal UtBF by 30–40% after 6–7 days in the absence of changes in arterial pressure or heart rate [22] and reduced responses to vasoconstrictors [20,23]. In addition, acute estrogen exposure starts to increase UtBF within 30–40 min, causing an even more robust (up to 10-fold maximum) rise in UtBF within 90–120 min after a bolus intravenous injection of 1–3 μg/kg E_2_β; this acute estrogen (1 μg/kg)-induced rise in UtBF declines gradually within 3–8 h after injection [24,25] and is independent of systemic responses [9]. The uterine vasodilatory effect of estrogens is of major physiological significance during the follicular phase of the ovarian cycle and pregnancy, in which circulating estrogens are significantly elevated [17,25,26,27]. During normal pregnancy, substantially increased endogenous estrogens are believed to be key players for upregulating UtBF, which provides the sole source of nutrients and oxygen supplies for the fetus and the exit of the metabolic wastes and respiratory gases of the fetus. UtBF is a critical rate-limiting factor for pregnancy health because (1) a dramatic rise in UtBF, especially in the last third of gestation, is directly linked to fetal growth and survival; and (2) insufficient rises in UtBF are associated with preeclampsia, intrauterine growth restriction (IUGR), and other pregnancy disorders, affecting the health of the infant after birth [28]. Notably, babies born to mothers with complicated pregnancies are predisposed to a significantly increased risk of developing cardiometabolic diseases, including cardiovascular diseases such as high blood pressure and coronary diseases, as well as type II diabetes [29]. From a nutrition standpoint, for supporting fetal development, constrained UtBF should be viewed as a “physiological” under-nutrition condition in fetal programming using the Barker hypothesis [30]. Thus, mechanistic investigations into estrogen-induced uterine dilatation during pregnancy will assist in the identification of therapeutic targets or even preventive options for pregnancy disorders.

The mechanisms by which estrogens regulate uterine hemodynamics during pregnancy are complex and not fully understood. Numerous physiological and pharmacological studies have accumulated compelling evidence demonstrating that an estrogen-induced rise in UtBF is associated with increased local uterine artery (UA) endothelial production of vasodilators such as prostacyclin [31], nitric oxide (NO) [32], carbon monoxide (CO) [33], endothelium-derived hyperpolarizing factor (EDHR) [34], etc. It has previously been shown that NO plays a more important role than prostacyclin in mediating the estrogen-induced rise in UtBF since the estrogen-induced rise in UtBF in nonpregnant sheep [9,32,35], the rise in UtBF in sheep during pregnancy [35], and the rise in the forearm blood flow in pregnant women [36] can be significantly attenuated by NO synthase inhibitors, but not by the prostaglandin synthase inhibitor indomethacin (8). Notably, NO inhibition maximally attenuates only ~79% of the estrogen-induced rise in UtBF [32] and modestly (~26%) inhibits baseline pregnancy-associated uterine vasodilation [35], clearly suggesting that additional mechanisms are involved. To this end, our novel work has recently identified that augmented hydrogen sulfide (H_2_S) production functions as a “new” UA vasodilator [37], which may offer great potential for uterine hemodynamic regulation. Nonetheless, our previous studies have shown that local infusion of the ER antagonist ICI 182, 780 can dose-dependently attenuate the estrogen-induced rise in UtBF in nonpregnant OVX and the pregnancy-associated rise in UtBF in intact pregnant ewes [25], suggesting that specific estrogen receptors (ERs) are involved in the estrogen regulation of uterine vasodilation in pregnancy.

The aim of this review is to summarize the literature on UA ERs with a focus on their roles in mediating the local UA production of vasodilators by estrogens and pregnancy and to deliberate on the potential clinic implications of dysregulated ER-mediated estrogen signaling in hypertensive pregnancy complications.

## 2. Classical ERs—ERα and ERβ in Uterine Vasodilation in Pregnancy

The biological functions of estrogens were initially identified to be mediated through the classical ERs, i.e., ERα and ERβ, which are ligand-dependent transcription factors. Both ERα and ERβ have been found in many species, including mammals (e.g., human [38], sheep [39], mouse [40] and rat [41]) and non-mammals such as chicken [42], xenopus [43], and fish [44]. ERα and ERβ are distinct proteins encoded by two distinct genes; in humans, the 66 kD ERα protein is encoded by *ESR1* (chromosome locus 6q25.1) and the 55 kD ERβ protein is encoded by *ESR2* (chromosome locus 14q23–24.1) [45,46]. The amino acid sequences of ERα and ERβ display a 59% sequence identity in their respective ligand binding domains (LBD), which represents a significant difference [47]. ERα and ERβ are expressed in a variety of tissues and cells, corresponding to the diverse biological effects of estrogens in numerous organs and cells throughout the body. In addition, ERα and ERβ show distinct expression patterns among different organs. ERα is predominantly expressed in pituitary, kidney, and both male and female reproductive systems such as the epididymis, testis, uterus, ovary, and breast, whereas ERβ is widely expressed in the reproductive system and brain [48]. The expression levels of ERα and ERβ are highest in the ovary and uterine endometrium, consistent with the fact that the female reproductive system is the primary target of estrogens [38,45].

There are also various variants of both ERs. ERα36 is a 36-kDa amino-terminal truncated product of the full-length ERα protein (ERα66), mainly located in the cell membrane and cytoplasm. ERα36 lacks the transactivation domain of ERα66 as well as the intrinsic transcriptional activity of estrogens, thereby competing with ERα66 to regulate the expression of genes with estrogen-responsive elements (EREs) in their promoter [49]. On the other hand, the overexpression of ERα66 suppresses the transcription of *ERα36*, which can be recovered by the forced expression of ERα36, independent of estrogens [50]. Moreover, ERα36 can mediate the estrogen activation of the extracellular signal-regulated kinases ½ (ERK½) [49] and phosphoinositide 3-kinases (PI3K)/protein kinase B (PKB/AKT) [51] pathways. ERα46 is another amino-terminal truncated product of ERα66 with a molecular weight of 46-kDa. ER46 has been identified in the vascular endothelial cells (EC) as a mediator of endothelial NO synthase (eNOS) activation in order to encourage the production of NO by estrogens [52]. Apart from ERα36 and ERα46, ERΔ7 is another recently reported 51-kD variant of ERα66 with the exclusion of exon 7; it acts as a dominant negative repressor of the uterotonic action in myometrium [53]. The physiological concentrations (100–700 pmol/L) of serum estrogens during the menstrual cycle can only bind to ERα46 and ERα66, but not to ERα36 [54], which can only bind with E_2_β at a relatively high kD of 2.2 nM compared to the physiological levels of serum estrogens [55]. Meanwhile, ERα36 can activate mitogen-activated protein kinase (MAPK) pathways in the presence of the inactively isomeric E2β 17α-estradiol [49] and testosterone [51] at the same concentration of E_2_β, suggesting that ERα36 is not the target for estrogenic actions in the physiological state, but merely acts as a membrane-associated ER. Nonetheless, whether this variant is expressed and has a functional role in the vasculature are currently unknown.

Other distinct ER subtypes have also been described. A novel ER subtype ERγ has been widely documented in fish species, including Atlantic croaker [56], zebrafish [57], and largemouth bass [58]. However, this ERγ has hitherto not been found in mammals. In addition, a brain ER isoform called ER-X has been described to be encoded by a distinct human gene [59]; however, this has not been found in other tissues and little is known regarding its function(s). Another isoform called putative ER (pER) has also been identified to possess distinct characteristics from that of ERα and ERβ [60]. pER has been reported in the ovary, but not any other reproductive tissues including the reproductive vasculature [60]. Thus, these ER subtypes are not discussed further in this review.

Uterine arteries, like all other large blood vessels, are comprised of a single concentric layer of EC that are surrounded by multilayers of smooth muscle cells (SMC). ERα and ERβ have been found in EC and SMC in various vascular beds, indicating that EC and SMC are the direct target cells of estrogens [61,62,63]. In addition, the membrane-bound G protein-coupled estrogen receptor 1 (GPER) has also been reported to be located on the plasma membrane, presumptively mediating the nongenomic effects of estrogens [64]. Thus, the vascular effects of estrogens are mediated by both “genomic” and “nongenomic” pathways. The presence of all three of these ERs has been documented in UA endothelial cells (UAEC) and UA smooth muscle cells (UASMC) in vivo [65,66,67]. Both ERα and ERβ and GPER are also expressed in various systemic arterial EC and SMC in vivo, including carotid, pulmonary, mesenteric, and renal arteries [68,69], and their expression levels are regulated in pregnancy in a vascular bed-specific manner [69]. We were the first to report both ERα and ERβ expressions in the UA endothelium and smooth muscles (SMs) in pregnant ewes ex vivo [39], which are retained in cultured primary ovine UAEC and UASMC in vitro. In that study, using immunoblotting with epitope-specific antibodies, we showed that a native ERα protein (67 kDa) is expressed in sheep UA, purified UA endothelium, and retained in UAEC in culture, similar to sheep ovary extracts (Figure 1A). However, we observed the presence of several ERβ protein bands in the extracts of these tissue samples and cells (Figure 1B), including the molecular weight of the 55-kDa band of the native full-length ERβ protein. Moreover, an additional ERβ band with a molecular weight of ~30 kDa was also detected in these samples. We named the 55-kDa band ERβ1 and the ~30-kDa band ERβ2 (Figure 1B). We also detected additional bands (Figure 1; indicated with *) with the specific anti-ERβ antibody, which could be other variants of ERβ (Figure 1B). Using RT-PCR and sequencing analysis, we confirmed these findings and showed that the ERβ2 protein represents a *N*-terminal truncated ERβ protein due to a 139-bp splicing deletion of exon 5 (Figure 1B,D–F). More recently, we also reported the functional roles of ERα and ERβ in cultured UASMC in vitro [70]. Collectively, these results indicate the presence of both ERα and ERβ proteins in sheep UA, UA endothelium and SM ex vivo, which are retained in cultured UAEC and UASMC in vitro.

In vivo, ICI 782, 180 dose-dependently inhibits exogenous and endogenous E_2_β-stimulated UtBF responses in nonpregnant OVX and intact ewes, respectively, demonstrating specific ER-mediated mechanisms for estrogens to cause uterine vasodilation. Moreover, we reported that ICI 780, 182 also dose-dependently attenuated baseline pregnancy-associated UtBF in ewes, showing a role of classical ERs in endogenous estrogen maintenance of uterine vasodilation during gestation [25]. Many ex vivo studies using wire myography recapitulate the rapid vasorelaxation effects of estrogens in uterine arteries and in selected systemic arteries such as mesenteric arteries [37]. Overall, these studies concluded that there was an involvement of classical ERs blocked by ICI 182, 780 mediating the E_2_β-induced relaxation of freshly prepared UA rings that have been pre-constricted by phenylephrine or other vasoconstrictors. Except for one early study [72], most have shown that pregnancy potentiates the vasorelaxant effects of estrogens including estetrol (E_4_) and E_2_β with relatively high potency in UA, which are sensitive to inhibition by ICI 182, 780 [25,73].

ER-mediated estrogen signing promotes uterine vasodilation [74]. In *ERα*-deficient mice, rapid estrogen-stimulated endothelial NO-dependent vasodilatation was significantly reduced [75,76]. In *ERβ*-deficient mice, estrogens even augmented phenylephrine-preconstricted dose-dependent vasoconstrictive effects in denuded aortic rings [77]. These data, obtained with ER knockout mouse models, show that ERα and ERβ may differentially mediate the responses of various vascular functions manifested by estrogens. ERα and ERβ often exhibit functional homogeneity and heterogeneity among different cells or tissues in mediating the effects of estrogens.

Specific ERα and ERβ agonists and antagonists are highly desirable and critically useful tools for discriminating the specific roles of ERα and ERβ in the complexity of estrogen signaling. Many synthetic compounds have been developed to preferentially bind ERα and ERβ. Specific agonists, propylpyrazoletriol (PPT) for ERα [78], and diarylpropionitrile (DPN) for ERβ [79], and specific antagonists, 1,3-bis(4-hydroxyphenyl)-4-methyl-5-[4-(2-piperidinylethoxy)phenol]-1*H*-pyrazole dihydrochloride (MPP) for ERα [80], 4-[2-phenyl-5,7-bis (trifluoromethyl)pyrazolo [1,5-a]pyrimidin-3yl]phenol (PHTPP) for ERβ [81], are pharmacological probes used frequently to define the specific roles of ERα and ERβ in mediating estrogen actions in vitro and in vivo.

Studies using specific ERα and ERβ agonists and antagonists have implicated clearly different roles of ERα and ERβ in the vasorelaxant effects of estrogens in the uterine and placental circulations. In human myometrial uterine arteries (arcuate/radial arteries), E_2_β, PPT (agonist for ERα), or DPN (agonist for ERβ) each induce comparable dose-dependent (1 nM–30 µM) relaxations that are pre-constricted by the thromboxane receptor agonist U46619 (1 µM). These findings suggest that both ERα and ERβ are involved in UA dilation in pregnant human uterine arteries. E_2_β can also relax human placental arteries, but with significantly lower potency than myometrial arteries. Although both ERα and ERβ are expressed in human placental arteries, it seems that E_2_β-induced placental artery relaxation is mediated by ERβ, because DPN can relax placental arteries, whereas PPT is without an effect [82]. Interestingly, the classical ER-mediated vasodilatory effects of estrogens are diminished in uterine and resistant mesentery arteries in chronologically aging rats and postmenopausal women [83]. Nonetheless, these findings suggest that ERα and ERβ differentially modulate the vascular tone in the uterine and placental arteries, highlighting that estrogens regulate human uterine and placental blood flows in a tissue-specific manner.

### Membrane ERs and Estrogen-Induced Uterine Vasodilation in Pregnancy

Following estrogen administration, UtBF begins to rise as early as 5–15 min, then reaches maximum levels around 90–120 min in various animal studies [2,9,20,21,24,84,85,86]. Nuclear ER-mediated estrogen signaling seems unable to explain this rapid vascular estrogenic response as *de novo* protein synthesis via nuclear ER-mediated gene transcription would take hours to take place. Moreover, cycloheximide completely abrogated the local estrogen-mediated rise in UtBF in the OVX ovine model [24]. Therefore, mechanistic studies have speculated that an estrogen-induced rapid rise in UtBF must be mediated by rapid estrogen signaling mediated by receptors localized on the plasma membrane. Indeed, estrogens can initiate rapid responses, such as calcium mobilization [87] and the generation of second messenger cyclic guanosine monophosphate (cGMP) [88] and cyclic adenosine 3′,5′-monophosphate (cAMP) [89] in various cells in vitro and in animals in vivo. Early mechanistic studies with the use of E_2_β-conjugated to bovine serum albumin (E_2_β-BSA) have shown that rapid estrogen signaling responses are mediated by classical ERs localized on the plasma membrane [90]. E_2_β-BSA is membrane impermeable and is widely used to demonstrate the presence of membrane ER, although free E_2_β is always a concern [91]. Nonetheless, there is solid evidence that both ERα and ERβ are present on the plasma membrane. In vascular EC, ERα has been shown to be partitioned into the specialized plasma microdomains, called caveolae, by interacting with caveolin-1 directly [71,92,93]. Both the plasma membrane-bound ERα and ERβ are responsible for the estrogen-stimulated rapid release of NO in UAEC in distinct ways [94]; however, the importance of this pathway in uterine vasodilation is unclear.

In 1997, a membrane receptor called G protein-coupled ER (GPR30/GPER) was initially cloned [95], which binds estrogens. The human *GPR30* gene is located at chromosome 7p22.3, which is composed of three exons in which the exon 3 contains its amino acid coding region. Interestingly, the region of the chromosome containing *GPR30* is thought to be related to familial hypertensive disease in humans [96,97]. GPR30 is an orphan receptor without known endogenous ligands; it has been proposed to be a bona fide membrane estrogen-binding receptor [97,98,99] and thus was renamed GPER by the International Union of Basic and Clinical Pharmacology [100]. GPER mRNA is widely expressed throughout the body, including in the lungs, liver, prostate, ovary, placenta, uterus and its vasculature [95,101,102,103], as well as ER-positive tumor cell lines [104]. Thus, it is not a surprise that GPER plays a ubiquitous role in the reproductive, nervous, endocrine, immune, and cardiovascular systems. Similar to classical ERs, GPER transcripts have been found in the vasculature throughout the cardiovascular system [105,106,107], including in the uterine arteries and veins [108]. Although both arteries and veins have shown GPER expression, only artery GPER expression is reduced by E_2_β [109], suggesting that local GPER might be fundamental in regulating estrogenic responses such as uterine vasodilation during pregnancy. However, *GPER* knockout has shown a minimal effect in regulating uterine growth by estrogens in mice [105,110,111,112]. We therefore suggest that GPER may function together with classical ERs in regulating uterine biology.

Although its cellular localization is still controversial, GPER has been regarded as a specific membrane estrogen receptor [98]. Although GPER interacts with E_2_β with estimated binding affinities of 3–6 nM [64], this is much lower compared with its binding affinities for classical ERs that are in the range of 0.1–1.0 nM [48]. It also interacts with anti-estrogens such as tamoxifen and the nonspecific ER antagonist ICI 782,180 [64], as well as many other estrogen receptor modulators [113], which makes it difficult to elucidate the specific role of GPER in target cells. Thus, the development of specific pharmacological tools, including the highly selective GPER agonist G-1 and antagonist G-15 [114], has greatly facilitated the characterization of GPER function. G-1 binds GPER with high affinity (Kd = 10 nM) without binding to ERα/β at concentrations as high as 10 μM [115], whereas G-15, with a similar structure as G-1, but lackingthe ethanone moiety that forms hydrogen bonds involved in receptor activation (103), displays a Kd > 10 μM for binding ERα/β [116].

Global *GPER*-knockout mice develop high blood pressure [112], atherosclerosis progression, and systemic inflammation [117], suggesting that GPER exerts cardioprotective effects. This conclusion is further supported by the fact that cardiomyocyte-specific *GPER* deletion results in an abnormal cardiac structure and impaired systolic and diastolic function in mice [118]. GPER may mediate the direct vasodilatory effect of estrogens in multiple vascular beds. Acute G-1 infusion decreases blood pressure in male rats, while the long-term injection of G-1 decreases mean arterial pressure in the hypertensive OVX female rats. These findings suggest that the activation of GPER potentially protects estrogen-deficient females from hypertension in rats [119]. In humans, G-1 dilates mammary arteries and, notably, G-1 dilates the precontracted human aorta and carotid artery more potently than E_2_β [120].

The first study attempting to specifically demonstrate the role of membrane ER-mediated signaling in uterine dilation was not successful in using wire myography to show a dilatory effect of E_2_β-BSA in freshly isolated nonpregnant and pregnant rat UA rings ex vivo [72]. However, GPER is expressed in the EC and SMC of mesenteric microvessels of female rats and was shown to mediate much of the E_2_β-induced dilation [121]. G-1 induces the dose-dependent relaxation of freshly prepared uterine radial artery rings isolated from both nonpregnant and pregnant rats that were pre-constricted with phenylephrine. Moreover, the expression levels of GPER protein were demonstrated to be greater in pregnant vs. nonpregnant radial arteries and the dilatory effect of G-1 in pregnant rat uterine radial arteries was more potent than that in nonpregnant ones [108]. These findings establish a casual role of GPER in pregnancy-dependent UA vasodilation in the rat model.

Although ERα has been shown to be localized on the plasma membrane of UAEC in in vitro culture [39,93], the importance of membrane-targeted ERα in estrogen-induced uterine dilation is yet to be fully determined. ERβ is also localized on the plasma membrane in other types of EC such as bovine aortic EC [122] and non-EC cancer cells [123]. In colon cancer cells, Galluzzo and her colleagues found that E_2_β could stimulate the association between ERβ and caveolae signature protein caveolin-1 [123]. Nonetheless, a recent study did not find direct ERβ targeting to the plasma membrane caveolae in UAEC [93]. Thus, it is highly likely that the role of ERβ in rapid uterine vasodilation to estrogen, if any, is an indirect mechanism that is yet to be determined.

## 3. Signaling Pathways of Estrogen Action

Estrogens signal via both “genomic” and “nongenomic” pathways in target cells expressing the classical ERs (ERα and ERβ) and the membrane-bound GPER. The genomic pathway is mediated by ERα/ERβ in the nucleus of target cells, where the ligated receptors function as ligand-activated transcription factors to regulate gene expression via direct interactions with estrogen-responsive elements (EREs) or indirect interactions with other response elements by binding with their respective transcription factors (e.g., specificity protein 1 (SP-1) and activator protein 1 (AP-1)) in the promoter of gaseous vasodilator synthesizing-enzyme genes such as *eNOS* and *cystathionine β-synthase (CBS)* [71,124] (Figure 2, route 1: nuclear actions).

Nonetheless, in some cell types, ERα and ERβ may play differential and, in some cases, even opposite roles in regulating cellular responses to estrogens [125]. In addition, estrogens can also signal via ER-independent routes by transactivating receptor tyrosine kinases (RTK) such as insulin-like growth factor 1 (IGF1) [126] and epidermal growth factor (EGF) [127] receptors (Figure 2, route 2: ER-independent manner).

As shown in Figure 2, route 3: non-nuclear actions, the nongenomic estrogen signaling is mediated by ERs (ERα and ERβ) localized at the plasma membrane [113,128], presumptively partitioned in the plasma membrane microdomain called caveolae via the binding between ERα and caveolin-1 in various cell types [90,92,93,123]. Nongenomic estrogen signaling is also mediated by the plasma membrane GPER. In this “extranuclear” mode, estrogen signaling is initiated in between seconds and minutes to activate downstream target proteins and elicit biological functions, presumably independent of nuclear events. Membrane ER-mediated nongenomic signaling can integrate with nuclear ER-mediated gene transcription via the protein kinase-mediated phosphorylation of ER and other ER-interacting transcription factors such as AP-1, SP-1, and cAMP response element-binding protein (CREB) to regulate latent gene expression in target cells in shown in Figure 2, route 4: cross-talk [129].

### 3.1. Nuclear ER Actions in UA

ERα and ERβ belong to the nuclear receptor superfamily principally consisting of three conserved functional domains, namely the *N*-terminal transactivation domain (NTD), the DNA-binding domain (DBD) and the ligand-binding domain (LBD) [130,131]. A conformational change occurs first, inducing receptor dimerization by the binding of E_2_β to ERα or ERβ in the cytoplasm [132]. When translocated into the nucleus, these E_2_β-bound ER dimers transactivate target genes that possess estrogen-responsive elements (EREs) within their promoters with a consensus sequence (5′-GGTCAnnnTGACC-3′) by interacting with DBD [133,134]; this transactivation is referred as “genomic” and was originally described for transcription factors. Besides promoters, EREs can also be located in the enhancer regions and/or the 3′ untranslated region of the target genes [135].

The genomic effects of estrogens via classical ERs provide the maintenance of vasodilation via the upregulation of key enzyme gene/protein expressions, which are regulated to keep the vasodilatation/vasoconstriction balance for the vascular tone [136]. Although both the homodimer and heterodimer of ERα and ERβ have been found to bind with the consensus palindromic ERE or half-palindromic ERE [137,138], the preference of the chromatin-binding spectrum does exist and has been found to be ER isoform-specific and ligand-dependent [139]. To further analyze this, the Grober lab generated the most significant position-specific probability matrix using multiple expectation maximization for motif elicitation (MEME) to make a comparison with the counterparts present in the transcription factor-binding profile database called JASPAR [140], and found that the ERE matrices derived from ERα selective, ERβ selective and ERα + ERβ-binding regions were identical, though ERβ failed to show ERE variant selectivity consequentially [140]. Although the consensus sequences of ERE share a high similarity, slight changes in the intrinsic sequence composition can alter the binding affinity for ER to ERE [141]. This moderately explains why the transcriptional roles of ERα and ERβ in mediating the same target of H_2_S-synthesizing enzyme gene expression to promote UA dilation are different for UAEC and UASMC [70,142].

Upon (or prior to) binding to ERE, ERs also show the ability to interact with other transcription factors such as SP-1 and AP-1, in a process referred to as transcription factor crosstalk, to initiate gene transcription independent of EREs [143,144,145,146]. One of the typical crosstalk events is facilitated by the binding of transcription factors to the GC-rich regions located in the promoter of the target gene with the presence of ERs [147]. For example, estrogens can stimulate eNOS [148] and cystathionine-γ-lyase (CSE) [149] expression by promoting the binding of SP-1 with its respective binding site in the promoter of the target gene. Another crosstalk event is functioned by a complex composed of ER and other transcription factors such as c-Fos, c-Jun and AP-1 [150,151]. In addition to the formation of a complex by protein–protein interaction between ERs and other transcription factors, the co-activator role (e.g., FoxA1) can also assist in creating accessible regions of the chromatin that ERα binds with to strengthen the physical interaction of ER with chromosome DNA [152,153].

### 3.2. Non-Nuclear ER Actions in UA

Since some estrogen-responsive genes do not have ERE-containing promoters, and intracellular responses of second messenger signal transductions to estrogen occur within minutes [154,155], the enzymatic pathways through the activation of membrane-associated ER are thought not to involve the direct ER activation of gene transcription and are referred to as “rapid” or “nonnuclear” ER signaling. Ligand binding by estrogens elicits conformational changes in ERs, which, in turn, are believed to interact with their neighboring signaling kinases (e.g., MAPK/ERK½, PI3K and Akt) leading to the activation of downstream cascades. Although both cell surface-bound ERα and ERβ exist in the human UA endothelium and no correspondence for ERβ has been found in caveolae [93], both ERα and ERβ may play regulatory roles in mediating nongenomic effects.

Since ERα and ERβ do not contain a trans-membrane domain [93] and are irrelevant with myristylation or prenylation [156], their ability to anchor signaling proteins at the endothelial membrane are mainly contributed by palmitoylation at the caveolae, the Ω-shaped invagination commonly present in EC [157]. However, most studies about the role of ER in caveolae focus on ERα associated with caveolin, the principal residual protein of caveolae, and eNOS [158,159]. In EC, caveolae-localized ERα activate PI3K, Akt and ERK½, subsequently enhancing the eNOS/NO pathway [90,160,161,162,163]; a similar study showed yjsy the pregnancy-specific changes in vasodilator production are associated with differences in Ca^2+^ and ERK½ signaling [164]. On the other hand, E_2_β stimulated the physical interaction between ERα and Gαi_2/3_ at the caveolae and activated the cGMP/protein kinase G type I β (PKG-Iβ) pathway, thereby leading to the rapid phosphorylation of CSE and enhanced H_2_S release [165]. Besides ERα, ER46, an *N*-terminal truncated ERα, can also form a complex with caveolin-1 in the caveolae located in EC to induce rapid eNOS phosphorylation and endothelial NO release [166]. Additionally, the role of ERβ in caveolae has only been confirmed in non-EC colon cancer cells [123]. The stimulatory effects of E_2_β in receptor de-palmitoylation are opposite between ERα and ERβ, in that de-palmitoylated ERα decreases the association with caveolin-1, whereas ERβ undergoes de-palmitoylation with receptor–caveolin-1 association increased. Then, the ERβ-caveolin-1 complex further activates p_38_^MAPK^ rather than Akt and ERK½ [123,167]. Nonetheless, our recent study in UAEC showed that ERβ is not localized in the caveolar subcellular microdomain [93], suggesting that ERβ may mediate the nongenomic estrogenic effects in a caveolin-1-independent manner. Interestingly, ER can be the direct targets for phosphorylation by protein kinases like MAPK, suggesting that the nongenomic estrogen signaling mediated through ER can, in turn, regulate ER itself via a feed-forward mechanism [168].

### 3.3. GPER-Mediated Nongenomic Effects in Uterine Vascular System

In addition to the classical ERs [169,170], some evidence has suggested that the effects of estrogen might be mediated by the transmembrane G-protein-coupled estrogen receptor (GPER). Coupled with the NO-cGMP signal pathway, GPER also appears to play a significant role in pregnancy-augmented vasodilatory effects in rat uterine arteries [108]. Because GPER is expressed in a vascular tone that responds to estrogen [106,171], and because of the antihypertensive response from local alterations in the renin–angiotensin system, a great deal of studies focuses on the participation of GPER in estrogen-mediated actions. Notably, GPER was found to activate acute signaling pathways including cAMP accumulation in vascular SMC as a vasorelaxant action [172]. In endometrial cancer cells, GPER was also shown to mediate estrogen signaling on the activation of PI3K/Akt and ERK½ pathways [173,174]. Since GPER could bind the ligands overlapping with ERα and ERβ, GPER-specific agonist G-1 [114] and the GPER-selective antagonist G-15 [116] were identified to unravel the functional roles of GPER, which shared a tetrahydro-3H-cyclopenta-[c]quinoline scaffold domain and showed an extremely high selectivity for GPER vs. ERα and ERβ. For instance, G-1 can mimic the cardioprotective effects of E_2_β in rats [175], and promotes the pregnancy-associated vasodilatory effects of estrogens in rat UA, which can be blocked by G-15 [108]. Like estrogens, G-1 can also activate ERK½ signaling in uterine leiomyoma and myometrial SMC; this stimulation was further inhibited by the MEK inhibitor PD98059 in myometrium SMC [176]. In vitro, E_2_β-induced GPER activation mediates estrogen-induced cell proliferation [174]. However, in *GPER* knockout mice, it is shown that GPER only plays a minimal role in mediating estrogen-induced uterine proliferation and hypertrophy [105,110,111,112]. Thus, in the uterus, GPER works together with the classical ERs to integrally regulate uterine physiological responses to estrogens.

Classical ERs and GPER proteins are expressed in numerous organ vasculature systems of the body tissues. As predicted, the crosstalk, coupled with multiple estrogen receptors, involve various pathways and notably change the final biological outcome. Apart from the direct mediation of estrogenic signal transduction at the cellular level, membrane-localized GPER also communicates with ERα at the membrane to bind with E_2_β and form a complex [177]. The ability of E_2_β to stimulate MAPK pathways via the interaction between ERα and GPER has been reported in endometrial cancer cells [174]. As a truncated form of ERα, ERα46 also interacts with ERα co-located in endothelial cell membranes and appears to be more effective in the modulation of nongenomic pathways, like enhancing eNOS phosphorylation, than ERα alone [52]. In vascular cells (e.g., EC [178] and SMC [155]) and non-vascular cells (e.g., uterine epithelial cells [179]), GPER can, in some cases, antagonize the effects of classical ERs. Nonetheless, E_2_β failed to specifically bind, and did not activate, cAMP, ERK½ or PI3K, or stimulate gene transcription in EC from a combination of ERα–ERβ knockout mice [128]. Thus, it remains to be determined whether GPER can act independently of ER to mediate rapid estrogen signaling responses. E_2_β could also activate calcium immobilization and PI3K/Akt signaling through the co-localization of endogenous ERα and endoplasmic reticulum GPER [97]. However, the mechanisms of recruitment of membrane-localized ER to the endoplasmic reticulum-localized GPER and the respective signal transduction are still less understood.

### 3.4. ER-Independent Estrogen Signaling and Estrogen-Independent ER Activation in UA

In some cells, estrogens can also mediate cell physiology independent of estrogen receptors. In the uterus of OVX mice, Richards et al. found that E_2_β treatment can directly enhance the tyrosine phosphorylation of insulin-like growth factor-1 receptor (IGF-1R), insulin receptor substrate 1 (IRS-1) and consequent PI3K signaling via the formation of an IGF-1R/IRS-1/PI3K complex [180,181]. E_2_β can also activate the epidermal growth factor receptor (EGFR) via the signaling of the lipid kinase sphingosine kinase-1 in breast cancer cells [182]. A great deal of studies also provided molecular evidence that the activation of receptor tyrosine kinases (RTK) (e.g., IGF-1R [126] and EGFR [127]) can trigger downstream MAPK and PI3K/Akt cascades, thereby indirectly activating ERs via the phosphorylation of Serine or Tyrosine residues [183,184,185].

Estrogen-independent ER activation has also been shown in vascular cells. Unliganded ERα inhibits EC proliferation but stimulates SM cell proliferation and activates the respective gene expression; this pattern is reversed in the presence of E_2_β [186]. ERs can also be activated in the absence of estrogens by growth factor receptor signaling coupled with the tyrosine kinase receptors of EGF and IGF [187]. In addition, fetal bovine serum (FBS) and EGF are able to activate ERα, in the absence of E_2_β, via a MAPK-independent pathway in vascular cells [188]. These studies suggested that the specific residues from ER itself or from other receptor ligands may be triggered in the absence of estrogens. Interestingly, the non-estrogen ligand-activated ER has been found to couple with signaling pathways different to those from the estrogen-activated ones, providing a hint that the effects of estrogens can determine ER-mediated intracellular mechanisms [189,190]. Whether these ligand-independent pathways for ER signaling pathways are involved in uterine hemodynamics in pregnancy is yet to be determined.

### 3.5. Posttranslational and Epigenetic Modification of ER Actions

Different types of posttranslational modifications, including phosphorylation, acetylation, sumoylation, and ubiquitination, etc., can directly regulate the stability and functions of ERs. As a nuclear transcription factor, modified ER transactivation will affect target gene expression [191]. For example, E_2_β enhances ER ubiquitination through degradation via the ubiquitin pathway in rat uteri [192]. Thus, in addition to the direct ER-regulated gene transcription, these posttranslational modifications may facilitate the transcriptional roles of ER in estrogenic responses in the vasculature. The direct p300-induced acetylation of ERα attenuates ligand-dependent transcription [193]. A conserved acetylation motif was identified in hinged region among mammalian species [194]; local site mutation results in increased cellular proliferation at sub-physiological levels of estrogens [195]. In addition, direct ligand-mediated sumoylation in ERα via the sumoylation site in its hinge region can regulate its transactivation function [191]. Sumoylation represses ERβ-mediated transcriptional activity, while preventing ERβ from ubiquitous degradation by competing with ubiquitin at the same phosphorylated sumoylation site. This mechanism is connected to the GSK3β-activated extension in response to the estrogens that act as sumoylation enhancers [196]. Although they are not the focus of this review, many other post-translational modifications, including phosphorylation, membrane targeting via palmitoylation, and interaction with caveolin-1, etc., have been reported to regulate ER function. However, the roles of these posttranslational mechanisms in mediating estrogen-induced uterine hemodynamics are yet to be explored.

In uterine endometrial cancer, abnormal methylation inactivates the ER gene and leads to subsequent hormone resistance [197]. Thus, ER-mediated gene expression may be also subject to epigenetic regulation in uterine vascular cells. Estrogens can also regulate microRNA (miRNA) expression, indicating that epidemic mechanisms for ER signaling may occur at the miRNA level. For example, E_2_β treatment upregulates miR-155, miR-429, and miR-451 and downregulates miR-181b and miR-204 in OVX mouse uteri [198]. In immature mice, E_2_β treatment is able to regulate miR-451, miR-155, miR-335-5p and miR-365 expression [199], suggesting that miRNAs can be potential biomarkers for estrogen responses in utero. Notably, most studies on the estrogen-induced ER-dependent miRNAs in uterine tissues have been identified from malignant carcinoma and primary cultured cells isolated from the umbilical cord, which are summarized and listed in Table 1. miRNA-mediated epigenetic mechanisms for regulating ligand-dependent ER signaling have gained more attention lately in the reproductive system [200]. For example, during pregnancy and labor, the miR-199a/214 cluster was found to mediate the opposing effects of estrogen and progesterone on uterine contractility [201]. miR-203 could mediate ER-regulated gene expression and physiological processes including cell proliferation and migration in rat uteri, as well as the etiology of endometrial carcinoma [202]. As summarized in Table 2, it is now clear that miRNAs are implicated in the normal uterine tissues, including its vasculature system, as well as in the malignant phenotype. In the UA, miR-210 plays an opposite role in uterine vasodilation during pregnancy by inhibiting large conductance Ca^2+^-activated potassium channels (BK_Ca_) channel–mediated relaxation and increasing pressure-dependent myogenic tone [203].

Currently, only a limited number of studies have reported on the epigenetic regulation of uterine hemodynamics during pregnancy. However, a critical role of methylation mechanisms in estrogen-induced uterine vasodilation has been demonstrated recently. In ovine UA ex vivo, Dasgupta et al. first reported that hypoxia induces the pregnancy-repressed methylation levels in CpG islands in SP-1-binding sites in the ERα promoter, thereby abrogating pregnancy-induced SP-1 binding to the ERα promoter [206]. This finding, in ovine UA, was followed up by Chen and colleagues, who found that hypoxia can decrease ERα expression, accompanied by increases in the methylation levels in the SP-1-binding site in the ERα promoter; the DNA methylation inhibitor 5-Aza-2′-deoxycytidin can restore the hypoxia-repressed ERα expression as well as the levels of SP-1 binding to the ER*α* promoter [207]. Pregnancy-triggered E_2_β- or progesterone-induced ERα expression can also be inhibited by hypoxia via enhanced methylation that curbs SP-1 binding to the ERα promoter. In this way, chronic hypoxia represses steroid hormone-upregulated BK_Ca_ channel activity in ovine UASMC, indicating a critical role of DNA methylation-affected ERα in relaxing UA SM via BK_Ca_ channels. Moreover, similar epigenetic mechanisms have also been reported in the ovine UA model, showing that hypoxia can abrogate pregnancy-induced demethylation in the SP-1-binding site as well as SP-1- and ERα-binding levels to the promoter of the BK_Ca_ channel assembly–pore forming auxiliary β1 subunit encoding gene *KCNMB1* [208]. In addition to direct regulation by environmental factors (e.g., hypoxia) or physiological states (e.g., pregnancy) of the CpG methylation level, the oxygen level and pregnancy-triggered steroid hormones can also indirectly alter the CpG methylation level by regulating DNA demethylation enzyme ten–eleven translocation (TET) methylcytosine dioxygenases in UA; E_2_β- or progesterone-induced TET1 expression can subsequently downregulate the methylation level in the SP-1-binding site in the *KCNMB1* promoter; TET1 enzymatic activity inhibition by its competitive inhibitor, fumarate, can restore the E_2_β- or progesterone-stimulated BK_Ca_ activity in UASMC [209]. These studies have demonstrated the role of DNA methylation in estrogen-induced uterine relaxation in the ovine UA.

## 4. Gasotransmitters in Estrogen-Induced Uterine Vasodilation in Pregnancy

As early as the late 1970s, RNA and protein synthesis inhibitors were used to determine if de novo protein synthesis is involved in estrogen-induced uterine vasodilation because UtBF rapidly increases at 5–15 min following a single bolus injection in various animals, including rabbits [216], guinea pigs [217], sows [218] and ewes [219]. Early studies showed that the unilateral infusion of cycloheximide significantly inhibits E_2_β-induced UtBF elevation during 90-min infusion, while the contralateral E_2_β-induced UtBF is unaffected; this inhibition lasts for more than 30 min after the removal of the cycloheximide infusion [24]. Follow-up studies have also shown that cycloheximide inhibited the estrogen-induced increase in uterine cGMP secretion in a dose-dependent manner, showing a 50% decrease in UtBF and a ~90% reduction in cGMP section at the time of the maximum UBF pattern [220]. These studies suggest that de novo protein synthesis is required for estrogen-induced uterine vasodilation.

The local infusion of ICI 180, 782 also dose-dependently inhibits estrogen-induced and pregnancy-associated uterine vasodilation [25]. In keeping with the rapid initiation of the rise in UtBF following estrogen administration [24,25], the mechanisms underlying the uterine vasodilatory effects of estrogens are expected to be mediated by ER-dependent and -independent rapid genomic and nongenomic pathways, as summarized in Figure 2. However, how each pathway is activated by estrogens and their relative significance is not fully elucidated. Nonetheless, there is ample evidence that estrogen-induced uterine vasodilation is mediated largely by the ER-dependent augmentation of the orchestrated production of vasodilators locally by the UA, including the family of “gasotransmitters”, i.e., NO, CO, and H_2_S (Figure 3). These gaseous signaling molecules are produced by EC and SMC, which coordinatively relax uterine arteries by activating cGMP-dependent pathways and potassium channels (K^+^) including BK_Ca_ and voltage-gated K^+^ (K_v_) channels, etc.

### 4.1. Nitric Oxide

NO is the most studied gaseous molecule of gasotransmitters that plays a role in a variety of biological processes in almost all types of organisms, ranging from single-cell organisms, to bacteria and plants, as well as animal and human cells. In the 1990s, the biology of NO exploded with the discovery that NO is an endothelium-derived relaxing factor (EDRF), which was discovered with the use of organ bath studies [221,222]. NO is biosynthesized endogenously via the conversion of l-arginine to l-citrulline in the presence of molecular O_2_ and nicotinamide adenine dinucleotide phosphate (NADPH) by a family of NO synthase (NOS) enzymes, including the Ca^2+^-dependent endothelial NOS (eNOS/NOS3) and neuronal NOS (nNOS/NOS2), and the Ca^2+^-independent inducible NOS (iNOS/NOS1) [223]. NO was the first gas molecule that was discovered to act as a chemical messenger in the body; endogenous NO, in a physiological concentration range, is essential for the normal functions of the brain, arteries, immune system, liver, pancreas, uterus, nerves and lungs, to name a few. These diverse functions have resulted in NO being named the Molecule of Year in 1992 [224]. In 1998, three scientists, Robert Furchgott, Louis Ignarro, and Ferid Murad, were awarded the Nobel Prize in Physiology or Medicine “for their discoveries concerning NO as a signaling molecule in the cardiovascular system” [225].

NO is highly reactive and can diffuse freely across membranes, and hence is ideal for a transient paracrine (between adjacent cells) and autocrine (within the cell producing it) signaling molecule. NO has a lifetime less than 10 seconds and the diffusion gradients of NO are limited by scavenging reactions involving hemoglobin [226], myoglobin [227] and other radicals [228]. However, NO can be stabilized in the blood and tissue by oxidation to nitrate (NO_2_^−^) and nitrite (NO_3_^−^), which can be transported to cells far away from where they were produced to function as endocrine molecules. NO also reacts with free thiols (-SH) in reactive cystines in proteins or metals to form nitrosothiols (-SNO) that function as NO reservoirs [229]. This reaction is called *S*-nitrosylation and it is reversible; thus, nitrosothiols have the potential to be converted back to NO under physiological and pathological conditions. Moreover, NO can react with superoxide (O_2_^−^) to form peroxynitrite (ONOO^−^) under oxidative stress conditions [230]; peroxynitrite is highly reactive with tyrosine residues, a reaction called protein *S*-nitration that normally results in the damage/deactivation of protein function [231].

In blood vessels, NO is mainly biosynthesized by eNOS, which is exclusively expressed in the EC (inner lining); endothelium-derived NO diffuses into the surrounding SMC, where it activates the soluble guanylyl cyclase (sGC)-cGMP pathway to relax the blood vessels, thus resulting in vasodilation and an increasing blood flow [232]. One of the main intracellular enzymatic targets of NO is sGC; in the presence of iron, the binding of NO to the heme region of sGC leads to the activation of the enzyme to produce the second messenger cGMP [233]. The major cellular targets of cGMP are cGMP-dependent protein kinases (PKGs), cGMP-gated cation channels, and cyclic nucleotide phosphodiesterases (PDEs), which break down cGMP [234]. In light of the diverse physiological functions of NO in organs throughout the human body, the targeting of some or all of these steps of NO signaling pathways has been a focus of development for various diseases. The most cited drug is sildenafil (Viagra), a common example of a drug that modulates a distal component of the NO pathway. Viagra does not stimulate the biosynthesis of NO, but enhances downstream signaling by shielding cGMP from degradation by cGMP-dependent phosphodiesterase type 5 inhibitor (PDE5) in the rat uterine arteries, thus promoting an increase in UtBF [235,236].

Magness and colleagues were the first to report that eNOS, but not nNOS, is expressed in the intact (but not the denuded) ovine UA and that pregnancy greatly stimulates eNOS expression ex vivo in the UA endothelium and in vitro in ovine UAEC [237]. This was supported by their subsequent study, demonstrating that acute E_2_β treatment can stimulate eNOS expression in the ovine UA endothelium and that chronic E_2_β perfusion is able to continuously increase eNOS expression in the UA endothelium [238]. Weiner and colleagues first reported, in guinea pigs, that pregnancy significantly increases Ca^2+^-dependent (but not Ca^2+^-independent) NOS activity (measured by ^3^H-Arginine to ^3^H-Citrulline conversion assay) in a variety of organs, including a twofold rise in heart, kidney, skeletal muscle, esophagus, and cerebellum, with the greatest being up to a fourfold rise in whole-UA homogenates, compared to nonpregnant controls [239]. Treatment with E_2_β (but not progesterone) also increases calcium-dependent NOS activity in these tissues, although they did not examine the effects of E_2_β on UA. Consistently, pregnancy and E_2_β both stimulate mRNA expression of eNOS and nNOS, but not iNOS, in the skeletal muscle. Moreover, the pregnancy-associated upregulation of Ca^2+^-dependent NOS activity is mediated by endogenous estrogens via classical ER, since it is blocked by the ER antagonist tamoxifen [239]. Subsequently, Magness et al. demonstrated similar results in ovine UA, showing that chelating Ca^2+^ with ethylene glycol-bis(β-aminoethyl ether)-*N*,*N*,*N*′,*N*′-tetraacetic acid (EGTA) downregulates NOS activity in ovine UA [237,240]. In the human model, Nelson and colleagues also showed that pregnancy-enhanced eNOS and nNOS expression accounts for the elevated Ca^2+^-dependent NOS activity in pregnancy [241]. Notably, baseline Ca^2+^-independent NOS activity was significantly higher in the pregnant than in nonpregnant UA, and a considerable amount of pregnancy-induced NOS activity is independent of Ca^2+^ modulation [241]. These findings in human UA were remarkably different to the counterpart study in guinea pig skeletal muscles.

Many groups have since examined the effects of pregnancy and estrogens on UA NOS–NO systems in many species including mice [242], rats [243], sheep [32,35,90], nonhuman primates [244], and women [245]. These studies have convincingly shown that (1) exogenous estrogens stimulate UA eNOS activation and NO production [90,238] and (2) the UA eNOS–NO system is significantly augmented during pregnancy and the follicular phase of the ovarian cycle [246,247], positively correlating with the elevated endogenous estrogens seen in these physiological states [241]. Moreover, with highly specific antibodies against each of the NOS isoforms becoming available, immunological studies, including Western blot analyses of mechanically purified endothelial proteins of ovine UA endothelium and SM and immunohistochemistry [238] or immunofluorescence microscopy [248], have concluded that pregnancy and estrogens specifically upregulate eNOS protein expression in the EC, but not SMC, in vivo in all species examined including women [241]. Consistently, some of these studies also have shown that blood and UA tissue levels of NO (measured by nitrite/nitrate) and cGMP are augmented during pregnancy and are also stimulated by estrogens [237,240,249].

Rosenfeld and colleagues performed a classical study using the OVX sheep model to determine the role of the local NOS–NO system in estrogen-induced uterine vasodilation. They reported that local UA infusion of the NOS inhibitor *N*(ω)-nitro-l-arginine methyl ester (l-NAME) dose-dependently inhibits the estrogen-induced maximal rise in UtBF in nonpregnant OVX ewes [35]. This study has established a critical role of local NO production in estrogen-induced uterine vasodilation, which is consistent with studies in nonpregnant ewes [32] and pregnant ewes [35], and in intact ewes in the follicular phase triggered by progesterone [25]. However, l-NAME can only inhibit at most ~79% of the estrogen-induced UtBF response in all these studies, suggesting that mechanisms in addition to NO exist to mediate estrogen-induced uterine vasodilation.

These physiological and pharmacological studies have promoted mechanistic studies for dissecting the cellular and molecular mechanisms controlling UA production of NO and how NO regulates uterine vascular functions further. As ex vivo studies have conclusively shown that local UA production is exclusively synthesized in the EC during pregnancy and upon estrogen stimulation and various other physiological regulators of UA vasodilation, such as vascular endothelial growth factor [250] and angiotensin-II [251], the Bird and Magness groups first developed novel primary culture models of UAEC from the main uterine arteries of nonpregnant and late (~130–140 days in ovine gestation, equivalent to the third trimester in human pregnancy) pregnant ewes [161] to study how UA local NO is synthesized during pregnancy and upon stimulation. Moreover, akin to nearly all primary EC culture models, the limited number of primary ovine UAEC directly isolated from the uterine arteries must be passaged so that there are enough cells for experimental investigation. Fortunately, after four to five passages, nonpregnant and pregnant UAEC maintain their physiological pregnancy-specific differences in baseline expressions of key enzymes for cell signaling and vasodilator (NO and prostacyclin) production, as well as their pregnancy-dependent responses to vascular endothelial growth factor (VEGF) [161]. These features make the ovine UAEC culture model the best available for dissecting the molecular mechanisms for UA hemodynamic regulation in pregnancy. Numerous studies from the Bird and Magness groups have shown that growth factors (VEGF, fibroblast growth factor 2 (FGF-2), and EGF) and signaling via MAPK and PI3K/Akt pathways in UAEC are programmed to stimulate eNOS activation and NO production via intracellular Ca^2+^ mobilization and Serine^1177^ phosphorylation in pregnancy [161]. Their studies have greatly advanced the cellular molecular mechanisms regulating the pregnancy-dependent upregulation of the local EC production of NO. Detailed information of these mechanisms can be found in their reviews [252,253,254].

We were the first to report UA ERα, ERβ mRNA and protein expression in EC and SMC in pregnant ovine UA ex vivo [39,71]; we also showed that ERα and ERβ are retained in UAEC cultures after up to five passages [71,90,255], showing that the ovine UAEC model is indeed suited to study estrogen-induced uterine vasodilation in vitro. Like the ovine fetal pulmonary artery EC [256], we showed that treatment with physiological relevant concentrations (1–10 nM) of E_2_β rapidly phosphorylates Raf-1/ERK½ within 2–5 min in pregnant ovine UAEC in culture; this pathway mediates E_2_β-induced NO production by indirect eNOS Ser^1177^ phosphorylation via a rapid nongenomic pathway with ERα localized on the plasma membrane [90]. We did not find that E_2_β mobilizes intracellular Ca^2+^ or activates PI3K–Akt in ovine UAEC [90], both of which are essential for rapid eNOS activation and direct Ser^1177^ phosphorylation, as reported for UAEC [94] and other ECs [257]. Nonetheless, the classical ER-mediated rapid nongenomic regulation of NO production in UAEC seems to be mediated by ERα because (1) ERα, but not ERβ, possesses binding domains to caveolin-1 [258]; (2) treatment with E_2_β promotes ERα membrane translocation to the caveolae [71,90]; and (3) rapid activation of eNOS involves the formation of a so-called steroid receptor fact action complex (SRFC) involving ERα, eNOS, and the enzymes that activate eNOS, such as AKT and ERK½, in the caveolae [122]. The membrane-bound GPER also mediates rapid estrogenic actions to increase pregnancy-dependent uterine vasodilation in rats, which is largely endothelium/NO dependent [108]. GPER has been shown to be present in rat UAEC [108]. However, the underlying mechanisms by which GPER mediates rapid eNOS activation by estrogens in UAEC are not currently known.

Concomitantly with the continuous increases in endogenous estrogens, UtBF increases continuously during gestation, essentially keeping pace with the growth rate of the fetus [259]. Thus, the production of local vasodilators such as NO must be subjected to chronical upregulation during pregnancy and stimulation by estrogens via genomic nuclear action to upregulate the expression of eNOS, a scenario as described above during pregnancy [237,260], the follicular phase of the ovarian cycle [238,261] and long-term estrogen replacement therapy (ERT) in OVX animals [22,262], or even in post-menopausal women [263]. In vivo, ERT selectively stimulates UA endothelial eNOS expression, potentially via upregulating eNOS transcription [22]. The eNOS gene promoter contains functional half-palindromic EREs and E_2_β stimulates ERα-dependent eNOS transcription in bovine pulmonary artery EC [264]. We reported that the activation of AP-1 may be important for UAEC eNOS transcription [265]. However, the specific roles of ERα and ERβ in the estrogen stimulation of eNOS transcription and expression in UAEC has not been determined. Nonetheless, our recent report has shown that treatment with PPT and DPN, similar to E_2_β, for 24–48 h, induces an increase in the stimulatory phosphorylation sites in ^Ser1177^eNOS and ^Ser635^eNOS, a decrease in the inhibitory phosphorylation site ^Thr495^eNOS, and an increase in total NO levels in ovine UAEC in vitro [94]. These findings suggest that the activation of both ERα and ERβ is involved in upregulating eNOS expression and activation to produce NO for mediating estrogen-induced uterine vasodilation in pregnancy.

Since estrogen treatment and pregnancy increases local UA production of both NO and cGMP, it is expected that the sGC–cGMP–PKG pathway is the major mechanism responsible for estrogen-induced uterine vasodilation in pregnancy. Indeed, myography studies have shown that the relaxation effects of pregnancy on pre-constricted UA rings are inhibited by the specific inhibitor of sGC [1-*H*-[1,2,4]oxadiazolo[4,3-a]quinoxalin-l-one (ODQ)] but recovered by the cGMP–hydrolysis enzyme PDE5 inhibitor sildenafil [236]. Since NO-induced activation of PKG directly phosphorylates BK_Ca_ channels [266], which causes vascular SM relaxation [232], the local infusion of the BK_Ca_ inhibitors tetraethylammonium chloride (TEA) and iberiotoxin (IBTX) can partially attenuate the estrogen-induced rise in UtBF in nonpregnant ewes and also inhibits baseline UtBF in pregnant ewes [267,268]. Overall, the available evidence shows a critical role of the sGC–cGMP–PKG–BK_Ca_ pathway in estrogen-induced uterine vasodilation in normal pregnancy. In addition, other K^+^ channels such as voltage-gated K^+^ (Kv) channels and small and medium conductance K^+^ channels [67], may be involved in estrogen-induced uterine vasodilation in pregnancy; however, these K^+^ channels seem to regulate uterine vasodilation under pathological conditions such as gestational diabetes [269].

NO is also involved in other uterine vascular changes adaptive to pregnancy such as UA remodeling [270]. We have shown that estrogens stimulate protein *S*-nitrosylation (SNO) in human umbilical cord vein EC (HUVEC) and ovine UAEC in vitro [255,271], in which ERα and ERβ, respectively, play different cell-specific roles [255]. Estrogen stimulation of endothelial protein SNO has been confirmed in rat UA in vivo [272]. The increased *S*-nitrosothiols stimulated by estrogens could serve as a NO reservoir for long-term NO signaling when it is needed [273]. We have reported that eNOS-derived endogenous NO, upon stimulation with estrogens and VEGF, stimulates SNO of a specific reactive cysteine (Cys80) among the four cysteines (Cys39/80/139/147) in cofilin-1 in EC [274,275]. Cofilin-1 is a major actin-binding protein that regulates cytoskeleton remodeling during cell proliferation and differentiation [276]. SNO results in the increased severing activity of cofilin-1, thereby regulating cytoskeleton remodeling and migration in response to stimulation with estrogens and VEGF [274,275]. Thus, it is speculative that increased SNO proteins may play a role in UA remodeling upon estrogen stimulation during pregnancy [270,277].

Numerous studies have demonstrated that mitochondrion is a major cellular organelle that is directly affected by endogenous NO synthesized upon estrogen stimulation [278,279,280]. NO exposure greatly affects the biogenesis of mitochondrion and its respiration chain [281]. Indeed, our previous studies have shown that total levels of mitochondrial SNO proteins are significantly increased by treatment with E_2_β to upregulate endogenous NO and by treatment with exogenous NO from NO donors in HUVEC. However, there are considerable differences (i.e., proteins and SNO levels) in the endogenous and exogenous NO-responsive endothelial SNO proteomes [282]. Interestingly, SNO of the mitochondrial protein is affected by locally synthesized NO since E_2_β-stimulated mitochondrial protein SNO in HUVEC is only enhanced by the overexpression of mitochondrion-targeting eNOS, but not by membrane-targeting eNOS [282], suggesting that estrogens affect mitochondrial SNO via local eNOS-synthesized NO.

### 4.2. Carbon Monoxide

CO is another biological gas that possesses similar physiological effects to NO in the vasculature. CO was initially regarded as a poisonous gas that is released with the formation of carboxyhemoglobin (COHb) under hypoxic conditions [283]. Under hypoxic conditions, high CO concentrations cause hypoxemia by competitive binding to the oxygen-binding sites of hemoglobin to form COHb, with an affinity approximately 245 times that of oxygen [284]. Thus, decreased concentrations of oxyhemoglobin (OHb) are inevitable in COHb-caused toxicity in hypoxemia [285]. In humans, prolonged or elevated CO exposure can cause a few acute clinical effects, including nausea, dizziness, and loss of consciousness. Symptoms of CO poisoning begin to appear at 20% COHb, while death occurs between 50 and 80% COHb [284].

In the 1950s, CO was found in the exhaled air in hospitalized patients [286]; however, this endogenously produced CO initially attracted little attention as an endogenous physiologic mediator until 1993, when Verma and colleagues identified a neurotransmitter function for endogenous CO in olfactory neurotransmission [287]. Subsequently, studies have shown that endogenously produced CO and low concentrations (e.g., 250 ppm) of exogenous CO possess various physiological functions, including a vascular role in regulating hepatic perfusion [288], anti-inflammation [289], anti-apoptosis [290,291], and the inhibition of vascular SMC proliferation [292], etc. Endogenous CO is produced physiologically via a single mechanism, the metabolism of heme to CO, biliverdin, and free iron by heme oxygenase (HO) [293]. HO exists as at least two isoforms that are encoded by two distinct genes, *HMOX1* and *HMOX2*. In humans, HO-1 is an inducible, low molecular weight heat shock protein-32 [294], whose expression is readily inducible by innumerable stimuli such as cAMP [295] and hypoxia [296]. On the contrary, HO-2 is a 36-kD protein that tends to be expressed constitutively and appears to be mainly regulated by steroids [297,298]. The highest levels of HO-2 are found in the brain, accounting for the majority of the neurotransmission functions of CO in the brain [298].

Several studies have established the role of the HO–CO system in pregnancy. Acevedo and colleagues have shown that the expression levels of HO-1 and HO-2 proteins are more than 10-fold greater (*p* < 0.001) in pregnant (regardless in labor or not in labor) than in nonpregnant human myometrium, which are upregulated by progesterone but not estrogens. Functional analysis also showed that HO-1 derived CO inhibits human myometrial contractility [299]. Follow-up studies have shown that the HO-1/CO system appears to be important in prenatal and postnatal development via regulating angiogenesis and immune hemostasis at the maternal–fetal interface. For mice lacking *Hmox1*, this is embryonically lethal [300], whereas a partial maternal *Hmox1* deficiency (*Hmox1*^+/−^) results in malformation of the maternal–fetal interface due to insufficient spiral artery remodeling and uterine natural killer (uNK) cell differentiation and maturation [301,302].

HO-1 is widely expressed in various tissues in the body and is highly inducible by oxidative stress, ultraviolet (UV) light, heavy metals and inflammation [303]; HO-2 has been found in vascular EC but is unresponsive to inducers of HO-1 [304]. Both HO-1 and HO-2 are expressed in vascular EC and SMC [304,305,306], and the HO-catalyzed formation of CO has been documented in blood vessels [307], suggesting that the vasodilatory roles of them might exist in the vasculature. Endogenously produced CO and exogenous CO can cause the endothelium-independent dilation of arteries and arterioles [294,308,309]. HO-1-produced endogenous CO appears to provide a tonic vasodepressor effect via the inhibition of an autonomic pressor mechanism [310]. In pulmonary arteries, endothelium-derived CO mediates acetylcholine (ACH)-induced vasorelaxation [311]. HO-2 protein is also highly expressed in large and small blood vessels and in adjacent astrocytes in the brain [312]. CO is a potent cerebral vessel dilator, as topically applied CO potently dilates piglet pial arterioles and the maximal response can be achieved by as low as 1 nM CO [313,314]. The vasodilatory effect of CO is mediated by the activation of sGC to generate cGMP in autocrine and paracrine fashions [294,308,309,311]. In vitro, CO can hyperpolarize vascular SM via modification of a histidine residue on the external membrane side of the BK_Ca_ channels [309,315]. In vivo, CO-induced cerebrovascular dilation was abolished by treatment with TEA and IBTX [313,314], suggesting that the vasodilatory effects of CO are mainly mediated by activating the sGC–cGMP–PKG–BK_Ca_ pathway.

Pregnancy upregulates HO-1 and HO-2 expression in human myometrial uterine blood vessels [299]. Although studies using partial Hmox1-deficient mouse models have shown that the HO-1/CO system regulates uterine spiral artery remodeling [301], there is no direct evidence regarding the role of the HO–CO system in estrogen-induced UA vasodilation during pregnancy. Nonetheless, physiologically relevant concentrations (0.1 nM) of E_2_β stimulate the ER-dependent upregulation of HO-2 mRNA/protein expression and CO production and elevate intracellular cGMP concentrations in HUVEC and HUAEC in vitro [33]. Moreover, equol (a non-steroid estrogen) can also stimulate ERβ (but not ERα)-dependent HO-1 expression via the PI3K/Akt–nuclear respiratory factor (Nrf) pathway in HUVEC [316]. These studies suggest the HO-1/CO system may play a role in uterine hemodynamics in response to estrogens in pregnancy. Moreover, low respiratory and serum CO are associated with hypertension in pregnancy and exhaled CO is inversely correlated with gestational hypertension in women [317], also providing indirect evidence for the role of the HO/CO system in uterine vascular adaptation to pregnancy.

### 4.3. Hydrogen Sulfide

H_2_S has been long recognized as a sewer gas with a characteristic foul odor of rotten eggs [318]. It was not until 1996 that Abe and Kimura first reported that endogenous H_2_S is produced by CBS in the hippocampus and that physiological concentrations of H_2_S selectively enhance *N*-methyl-d-aspartate (NMDA) receptor-mediated responses and facilitate the induction of long-term hippocampal potentiation, establishing that endogenous H_2_S functions as a neuromodulator in the brain [319]. Since then, research into H_2_S biology and medicine has flourished and demonstrated a plethora of biological, physiological, and pathological functions of H_2_S in the neuronal, immune, cardiovascular, endocrine and digestive systems, and cancers [320]. Due to similar biochemical properties and functions, as well as mechanisms of action, H_2_S has now been widely accepted as the third “gasotransmitter” after NO and CO [321].

Endogenous H_2_S is mainly synthesized from l-cysteine by the two pyridoxal-5′-phosphate-dependent enzymes CBS and CSE of the trans-sulfuration pathway [322,323]. CBS catalyzes the transfer from serine and cysteine to cystathionine, during which H_2_S is produced as a product. CSE synthesizes H_2_S through three pathways, including (1) the β-elimination of cysteine to pyruvate, H_2_S, and NH_4_^+^; (2) the γ-elimination of homocysteine to 2-ketobutyrate, H_2_S, and NH_4_^+^; and (3) the β- or γ-replacement reaction between two cysteine or two homocysteine molecules, with lanthionine or homolanthionine as the co-products, respectively [324,325]. CBS and CSE expression can be tissue/cell specific, as both are needed to generate H_2_S in some tissues, while one enzyme is sufficient in others [326,327]. These enzymes are found in numerous tissues in mammals [320]. In addition, 3-mercaptopyruvate sulfurtransferase (3-MST), cytosolic cysteine aminotransferase (cCAT) and mitochondrial cysteine aminotransferase (mCAT) can also produce H_2_S, but to a lesser extent [328]. CBS and CSE are expressed all over the body; however, CBS plays a major role in H_2_S biosynthesis in the central nervous system whilst CSE is the main H_2_S-synthesizing enzyme in the vasculature [329]. As the final product in cysteine metabolism catalyzed by CBS and CSE, however, H_2_S generates negative feedback by inhibiting the bioactivities of CBS and CSE [330].

As a biological gas, H_2_S can freely cross cell membranes to act as a signaling molecule to regulate numerous biological pathways, such as Ca^2+^ signaling, apoptosis, redox signaling, angiogenesis, vasodilation, and cardioprotection, etc. [331,332]. Under physiological conditions, the concentrations of H_2_S in the human body are low since, once synthesized, it will be easily metabolized and oxidized in the mitochondria. In vivo, H_2_S is oxidized to persulfide sulfur by sulfide:quinone oxidoreductase (SQR) or to thiosulfate (SSO_3_^2−^) and sulfate (SO_4_^2−^) by persulfide dioxygenase, rhodanese and/or sulfite oxidase [333]. This process seems to be important in oxygen (O_2_)-sensing cells and tissues including systemic and uterine vasculatures where H_2_S has been proposed as an “oxygen sensor”, which mediates the tissue response to hypoxia [334].

In recent years, *S*-sulfhydration or *S*-persulfidation has been identified to be the primary signaling mechanism for mediating H_2_S action. Sulfhydration denotes the formation of -S-SH adducts in proteins, concerting free thiols (-SH) to persulfide (-SSH) groups in the reactive cysteines in proteins. Sulfhydration normally results in the increased reactivity of the modified cysteines due to the increased nucleophilicity of -SSH compared with -SH [335]. Since cysteine is an essential amino acid, this post-translational modification at reactive cysteines affects the entire proteome and thus inevitably participates in nearly all biological pathways. Malfunctions in this critical cellular process have been implicated as causal factors in numerous diseases, including preeclampsia [336,337,338,339,340]. This is testament to the importance of sulfhydration as a signaling process and explains the growing interest in understanding the regulation and mechanisms of the process. Indeed, there are many proteins that have been identified as sulfhydrated proteins. In some vessels, H_2_O_2_ is an endothelium-derived hyperpolarization factor (EDHF) whose function is mediated by oxidizing Cys^42^ in PKG to induce disulfide-linked PKG homodimerization and activation [341,342]. Mice with Cys^42^ in PKG replaced by serine are hypertensive and exhibit impaired vasorelaxation response to NaHS in their resistant vessels, suggesting that polysulfides derived from NaHS can stimulate PKG [343]. H_2_S increases the *S*-sulfhydration of mitogen-activated protein kinase 1 (MEK1) in HUVEC, while mice lacking the *CSE* gene have lower sulfhydrated MEK1 [344]. Various ion channels are also target proteins that can be sulfhydrated by H_2_S, resulting in either increased activity, such as in voltage-activated Ca^2+^ channels in neurons [345,346,347] and Cl^−^/HCO_3_^−^ exchangers in vascular SMC [348], or decreased activity, such as in l-type Ca^2+^ channels in SMC [349,350].

CSE-derived endogenous H_2_S is a physiological vasorelaxant, as global *CSE* knockout mice display pronounced hypertension and diminished endothelium-dependent vasorelaxation in association with markedly reduced H_2_S levels in the serum, heart, aorta, and other tissues [351]. H_2_S can relax freshly prepared blood vessel rings from many vascular beds, including in cerebral [314], mesentery [352], coronary [353], uterine [37], and placental [338] arteries. However, unlike endogenous NO, which is exclusively produced by EC, endogenous H_2_S in the vascular wall is synthesized by both EC and SMC and, collectively, they regulate vascular tone and homeostasis. The vasodilatory effects of H_2_S have been shown to be mediated by the activation of the adenosine triphosphate (ATP)-sensitive K^+^-channel (K_ATP_) and BK_Ca_ channels, resulting in SM relaxation [331,351,354]. In rat aortas, endogenous H_2_S maintains whole-cell K_ATP_ currents, while exogenous H_2_S activates K_ATP_ channels by increasing the availability of single channels of this type [355]. In freshly prepared endothelium-intact and denuded rat mesentery artery rings, exogenous H_2_S from donors (100 μM NaSH) potently induces vascular relaxation, which is partially attenuated by the mixture of the endothelial small conductance K^+^ channel (SK_Ca_) inhibitor apamin, and intermediate conductance K^+^-channel (IK_Ca_) inhibitor charybdotoxin, or the K_ATP_ channel inhibitor glibenclamide [337]. However, the dilatory effects of H_2_S in rat mesentery arteries are not altered by the sGC inhibitor ODQ or the PKG inhibitor KT5823 [337]. These studies demonstrate that exogenous H_2_S activates K^+^ channels including endothelial IK_Ca_, SK_Ca_ and K_ATP_, but not sGC and PKG, which are important for mediating the vasodilatory effects of H_2_S. These studies also suggest that activation of K_ATP_ channels by H_2_S is independent of the sGC–cGMP pathway, although H_2_S can increase cGMP to activate PKG in EC [356].

Instead, H_2_S-induced vasorelaxation has been shown to be largely mediated by the cGMP–PKG-independent sulfhydration of various ion channels in vascular SM. For example, treatment with a H_2_S donor, Na_2_S, dilates vessels via sulfhydrating the putative proangiogenic Ca^2+^-permeable transient receptor potential (TRP) V4 channels, resulting in a Ca^2+^ and K^+^ influx in rat aortic artery ECs [357]. H_2_S also sulfhydrates Kir 6.1 [337] and sulfonylurea 2B [358] in K_ATP_ channels and K_V_4.3 in K_V_ channels [359] in SMC to promote SM relaxation. Interestingly, pregnancy stimulates the expression and activation of various K^+^ channels, including BK_Ca_ and K_ATP_ in the UA [360,361,362], most likely via the classical NO–cGMP–PKG and PKC pathways [363].

H_2_S is also a potent pro-angiogenic factor because the pharmacological inhibition of CSE and CBS bioactivity or gene silencing using *CBS* or *CSE* siRNAs results in reduced angiogenesis in multiple in vivo angiogenesis models [356,364] and exogenous H_2_S also stimulates angiogenesis in vitro and in vivo [356,365,366]. The angiogenic activity of H_2_S is mediated by the activation of the cGMP pathway as ODQ and KT5823 can partially block H_2_S-induced angiogenesis [356,364]. Moreover, exogenous and endogenous H_2_S activates EC eNOS/NO during in vitro and in vivo angiogenesis [356,367]. Endothelial eNOS-derived NO is a physiological angiogenic factor [368]. These studies suggest that H_2_S regulates angiogenesis by interacting with eNOS/NO. Additionally, exogenous and endogenous H_2_S has been shown to induce apoptosis and inhibit proliferation in human and rat aortic SMC in vitro via the activation of MAPK pathways [369,370,371]. 

Whether or not the H_2_S system was present in the UA was not known until 2015, after we first reported that exogenous E_2_β stimulates UA H_2_S production via the selective upregulation of CBS expression in OVX nonpregnant (NP) ewes [372]. In that study, we first posited the novel role of H_2_S in the pregnancy-associated rise in UtBF because of the potent vasodilatory effects of H_2_S, as discussed above. Indeed, we were recently the first to report that UA H_2_S production is significantly augmented with a ~nine-fold increase in EC/SM CBS protein, but not CSE and other enzymes (3MST and CAT), in pregnant, but not nonpregnant, ewes [373] in vivo and in women [37] ex vivo. We also reported that a slow-releasing H_2_S donor GYY4237 dose-dependently stimulates relaxation of phenylephrine-preconstricted UA rings isolated from both pregnant and nonpregnant rats, but with significantly greater potency in the pregnant state (Figure 4). In addition, GYY4237 does not relax the mesentery artery in pregnant rats [37]. Thus, our findings show that H_2_S stimulates pregnancy-dependent UA vasodilation, establishing H_2_S’s function as a “new” UA vasodilator.

Our recent work has shown that UAEC and UASMC H_2_S biosynthesis is stimulated by exogenous estrogens [372] and is also augmented during ovine [373] and human pregnancy [37], correlating with endogenous estrogens. Notably, exogenous and endogenous estrogens stimulate UA H_2_S biosynthesis via the selective upregulation of CBS but not CSE mRNA and protein expressions in EC and SMC ex vivo [37,372,373]. However, unlike systemic arteries in which CSE seems to be the enzyme for H_2_S production [329], our data consistently show that CBS is the major enzyme responsible for augmenting UA H_2_S biosynthesis because other H_2_S producing enzymes, i.e., CSE, 3MST, mCAT, and cCAT, are not altered in human UA in the menstrual cycle and during pregnancy [37].

We have further studied the mechanisms controlling UA H_2_S biosynthesis upon estrogen stimulation. In a similar manner to in vivo conditions, treatment with E_2_β increases CBS mRNA and protein in both ovine UAEC [142] and UASMC [70] models in a time- and concentration-dependent manner in vitro, suggesting that the UA *CBS* gene may be regulated by estrogens and by transcriptional mechanisms during pregnancy. The human *CBS* promoter contains multiple putative cis elements for binding various transcription factors, including ERE and binding sites for SP-1, AP-1, and AP-3 [374,375]; when transfected into both UAEC and UASMC, treatment with E_2_β results in the transactivation of the human *CBS* promoter in both cell models [70,142]. Notably, treatment with PPT, DPN, or their combination, stimulates CBS mRNA and protein expressions to levels similar to that of E_2_β-induced responses in both ovine UAEC and UASMC. On the other hand, co-treatment with MPP, PHTPP, or both, completely blocked E_2_β-induced CBS expression. These findings show that the activation of either ERα or ERβ suffices to mediate *CBS* transcription in response to estrogens on the UA wall.

Treatment with E_2_β also stimulates CSE mRNA and protein expression via ERα and/or ERβ-dependent transcription in ovine UAEC and UASMC in vitro [70,142]. These data demonstrate that *CSE* is also an estrogen-responsive gene in UAEC and UASMC in vitro. Although the *CSE* promoter also contains EREs [374,375], the estrogen stimulation of CSE expression in UAEC and UASMC in vitro is unexpected because it contradicts not only our ex vivo studies showing that exogenous or endogenous E_2_β does not alter CSE mRNA and protein in UA endothelium in women [37] ex vivo and sheep [372,373] in vivo and ex vivo, but also other studies showing that E_2_β does not stimulate CSE expression in mouse mesenteric SMC in vitro [376]. The discrepancy between the effect of estrogens on CSE expression in UA in vitro and ex vivo is currently unknown, but may, in part, result from the loss of cell–cell interactions and the microenvironment in which EC and SMC reside ex vivo. Nonetheless, these studies urge that caution should be exercised when interpreting in vitro findings as they pertain to ex vivo conditions, because they can sometimes produce different outcomes. We have developed novel human UAEC and UASMC models from pregnant and nonpregnant women for uterine hemodynamics research. In these cells, we found that pregnancy augments VEGF-stimulated H_2_S production by the selective upregulation of CBS expression, without altering CSE expression in vitro [250]. Primary human UAEC retains the pregnancy-dependent expression of ERα and ERβ in culture. Treatment with E_2_β dose-dependently stimulates angiotensin A2 receptor (AT_2_R) expression in human UAEC prepared from pregnant (but not nonpregnant) women, which is mediated by ERβ interactions with specific EREs in the human AT_2_R promoter [377]. These studies suggest species differences in UAEC and UASMC in response to estrogens in vitro.

More recently, it was shown that E_2_β can rapidly stimulate H_2_S release within minutes in hUAEC in vitro. This rapid H_2_S release in response to estrogens is mediated by ERα interaction with the Gα subunit Gαi-2/3 on the plasma membrane, resulting in the transactivation of *particulate guanylate cyclase*-*A* (*pGC-A*) for generating cGMP, thereby activating PKG-I; activated PKG-I then phosphorylates CSE to increase H_2_S production. The silencing of either *CSE* or *pGC-A* in blunted estrogen-induced aorta vasodilation in mice suggests that this nongenomic pathway may plays a role in estrogen-induced vasodilation via endothelial CSE-H_2_S [165]. CBS can also be phosphorylated on serine^227^ directly by PKG, resulting in the action of the enzyme and thereby increasing H_2_S production [378]. However, whether GPER or ERβ also play a role in the nongenomic activation of CSE and whether CBS has a role in rapid H_2_S production in response to estrogens are unknown. It is also not known whether nongenomic pathways of estrogen signaling also regulate (in UA) H_2_S production.

Nonetheless, it is important to point out that research on H_2_S in uterine hemodynamic regulation during pregnancy is still in its early stages in comparison to NO and CO. There is much to be learned before a definite physiological role of endogenous H_2_S in estrogen-induced and pregnancy-associated uterine vasodilation in pregnancy can be established for exploring the therapeutic potential of H_2_S in pregnancy complications.

### 4.4. H_2_S Interactions with NO and CO in the Vasculature

NO, CO, and H_2_S are formed in the UA endothelium and/or SM in the vascular wall. With a half-life between seconds (NO) and minutes (CO and H_2_S), each can activate specific pathways to regulate vascular tone and homeostasis independently. An obviously question is, among the three, what is the relative significance in regulating vascular tone and homeostasis? The available evidence, however, cannot distinguish their relative significance from one another, but rather supports an idea that the three function coordinatively in the cardiovascular system: for example, as discussed above, H_2_S and NO stimulate distinct and common pathways to elicit their angiogenic and vasodilatory effects. However, H_2_S and NO stimulate angiogenesis in vivo in a mutually exclusive manner [379]. In addition, H_2_S-induced rat aortic artery vasorelaxation can be partially attenuated by either endothelium removal or the application of l-NAME to endothelium-intact rings [331], suggesting that the dilatory effects of H_2_S on the aortic artery are partially endothelium/NO-dependent. In this context, it is proposed that an intact endothelium might serve as a physical buffer system to retain H_2_S in the vessel wall so that its vasorelaxant effect can be potentiated and prolonged [331]. However, endogenous H_2_S or H_2_S donors stimulate eNOS phosphorylation and NO production in EC [356,367], suggesting that H_2_S functions upstream of the eNOS–NO pathway in the vasculature. Although most studies have shown that H_2_S positively cooperates with NO in the vascular system, the concentrations of H_2_S donors used are relatively high in these studies. In contrast, less than 100 µM NaHS is shown to be paradoxically vasoconstrictive by suppressing cAMP production in vascular SMC, and to reverse vasorelaxation induced by endothelium-derived vasodilators such as acetylcholine and histamine; this contractile activity of H_2_S is attributed to the direct inhibition of eNOS [380]. In addition, high concentrations of H_2_S, like NO and CO, are cytotoxic and induce cell death. Thus, the biological functions of all three depend on a U-shaped local concentration-dependent manner.

The interplay between H_2_S and NO might not exert enhancement or potentiation effects. Ali et al. showed that NaSH treatment can diminish the vasorelaxant effects induced by each NO donor [sodium nitroprusside (SNP), morpholinosidonimine (SIN-1) and *S*-nitroso-*N*-acetylpenicillamine (SNAP)] in rat aortas [381], suggesting that the interaction of H_2_S and NO may form a product that does not respond to vasorelaxant effects. The product of interaction has been characterized by the fact that SNP prevents the relaxation effects of H_2_S by the generation of nitroxyl (HNO) [382]. H_2_S conditionally reacts with *S*-nitrosothiols, resulting in the formation of thionitrous acid, the smallest *S*-nitrosothiol (HSNO), which serves as a NO, NO^+^, and HNO donor, depending on the reaction condition [383]. Cortese-Krott et al. first demonstrated that the short-lived HSNO has the potential to react with sulfide, forming multiple bioactive products, including the nitrosopersulfide SSNO^−^, which can sustainably generate NO and polysulfides on decomposition, accompanied by the nitrosothiol-induced stimulation of sGC signaling [384]. To display the sustainable SSNO^−^ effects via NO/H_2_S interplay, Berenyiova et al. recently prepared the product “SSNO^−^ mix” via the interaction between Na_2_S and the NO donor *S*-nitrosoglutathione (GSNO) based on Cortese-Krott’s study [385]. In rat uteri, GSNO showed a weaker relaxation effect than sulfide on uterine contractility; the interaction of sulfide/GSNO could diminish the relaxation effects, even at a 1:10 ratio [385]. Meanwhile, GSNO relaxed the pre-restricted rat aortic rings with a stronger and faster effect than sulfide, but a weaker one than “SSNO^−^ mix”. This study on sulfide, GSNO and the reaction products of the sulfide/GSNO interaction has indicated that their effects on uterine and vascular SM are different, and the products of the sulfide/GSNO interaction can account for the sGC activation [385]. However, the vasorelaxation effects of the products formed from sulfide/GSNO interaction are yet to be determined in the uterine vasculature.

CO and NO share many similar functions, as they activate common downstream signaling pathways. In addition, HO proteins share a high homology in the C-terminal amino acid sequence with NOS [386]. In response to low-concentration CO stimulation, vascular EC and blood platelets release NO and generate peroxynitrite [387]. In the aortic artery endothelium, HO-1 and HO-2 concomitantly express with eNOS, collaboratively contributing to vasodilation [388]. This suggests that the HO/CO and eNOS/NO systems also interact with each other in the vascular system. However, the functions of CO and NO can be different. For example, CO stimulates sGC activity with a low tissue NO level, whereas CO inhibits sGC activity with a high tissue NO level [389]. Thus, the potential interplays between CO and NO also occur in different manners, being synergistic or antagonistic, and dependent on local concentrations.

Most studies regarding interactions among gasotransmitters are in the context of NO and H_2_S and NO and CO. However, the interaction between CO and H_2_S is very understudied. Exogenous H_2_S is able to increase HO-1 expression and CO production, thereby alleviating the elevation of pulmonary arterial pressure in a chronic hypoxic animal model [390], suggesting that H_2_S and CO interact in pulmonary SMC to regulate blood pressure. However, in cultured aortic SMC, the blockage of endogenous CO by a HO-1 inhibitor zinc protoporphyrin (ZnPP) decreased CSE expression and H_2_S production, while the inhibition of CSE by propargylglycine (PAG) increased HO-1 expression and CO production, but treatment with NaSH (100 µM) decreased HO-1 expression [391]. These results suggest that endogenous H_2_S/CSE and CO/HO systems seem to antagonize one another in aortic SMC in vitro.

The interactions among NO, CO, and H_2_S certainly provide another dimension to the complexity of gasotransmitter signaling and function in the vascular system. All three have been shown to play sometimes redundant or important roles in the uterine vasculature, but their interactions in regulating uterine hemodynamics are unknown. However, it is expected that their complimentary interactions are important since, when one system is dysfunctional, it may be compensated by the other(s) [392].

## 5. Estrogens and Gasotransmitters in Preeclampsia

### 5.1. Pathophysiological Evidence

Preeclampsia (PE) is a pregnancy disorder characterized by blood pressure greater than 140/90 mmHg and proteinuria that weighs more than 0.3 g per day after the 20th week of gestation in previously normotensive and nonproteinuric pregnant women [393,394]. It affects 2–8% of pregnant women in developed countries and 15–20% of all pregnancies in developing countries [395,396]. Preeclampsia is a major pregnancy complication that increases maternal and perinatal mortality and morbidity. The pathogenesis of PE is incompletely understood. Although there are many different phenotypes of PE, the common denominator is the placenta, as the clinical manifestations disappear within a few days after the delivery of the placenta [397]. It is generally believed that shallow trophoblast invasion and impaired uterine spiral artery remodeling during placentation results in decreased UtBF to the placenta; ischemia/hypoxia further stimulates placenta production of harmful factors such as inflammatory cytokines [i.e., tumor necrosis factor α (TNFα) and interleukin 6 (IL-6)] and antiangiogenic factors (i.e., soluble fms-like tyrosine kinase (sFlt-1), and soluble endoglin), which further cause systemic inflammation and endothelial dysfunction [398,399].

Circulating estrogen levels have, in general, been found to be decreased in preeclamptic women compared normotensive women during pregnancy [400]. Low estrogen levels during pregnancy are also associated with insufficient UtBF to the gravid uterus as well as pregnancy complications such as preterm delivery, intrauterine fetal growth restriction, and PE [401,402]. During pregnancy, estrogens are primarily produced by the placenta [15] and they promote angiogenesis and vasodilation [403]; ischemia/hypoxia-induced aromatase deficiency to the placenta can result in decreased estrogen production in PE [404]. With more reliable methods for accessing estrogens and their metabolites, studies have further validated a close link between estrogen dysregulation and PE [402]. Mice deficient of catechol-*O*-methyltransferase, which converts 17β-triol to one of its major metabolites, 2-methoxyoestradiol, develop PE-like symptoms [405], further supporting the causal role of reduced estrogens in PE.

Although measurements of NO and its metabolites (nitrite and nitrate) in pregnancy have resulted in conflicting results [406,407,408], there is a consensus that NO bioavailability is significantly decreased because endothelial NO production is reduced, as dysfunctional EC is a hallmark of PE [254,409] or because eNOS is uncoupled to produce superoxide (O_2_^−^) [410]. In preeclamptic women, decreased NO bioavailability and increased O_2_^−^ result in the formation of ONOO^−^, which is a potent cytotoxic anion that interacts with tyrosine to decrease protein function via nitrosation [411], resulting in further placental damage, EC dysfunction, and inflammation in PE. In addition, reduced NO lowers UtBF, which results in the impairment of shear stress-mediated NO release and diminished UA remodeling [412]. This mechanism is supported by the NOS antagonist l-NAME-inhibited matrix metalloproteinase (MMP)/metallopeptidase inhibitor (TIMP) pathway in pregnant rats, with increased arterial wall collagen and elastin content and impaired expansive remodeling compared to normal pregnancy [277].

HO-2 expression on syncytiotrophoblast is highest in early human pregnancy and reduces toward term [413]. However, placental villous endothelial HO-2 expression is notably reduced in PE and IUGR pregnancies [414]. Indeed, the end tidal CO (ETCO) levels are significantly lower in PE than gestationally hypertensive and normal pregnant women [317]. Chronic exposure to CO at 250 ppm dilates UA and UtBF in mice [415]; this may explain the lower incidence of PE in pregnant women who smoke, since cigarette smoking releases increased CO levels in the maternal blood [415]. These observations suggest that the HO/CO system is important for uterine hemodynamics regulation during normal pregnancy and that its dysfunction is a causal factor for PE.

Genetic *CBS* deficiency results in homocysteinemia, causing endothelial dysfunction and hypertension in both animals and humans [416]. *CSE* knockout mice display reduced serum H_2_S levels, accompanied by hypertension and diminished vasodilation [351]. These observations have provided direct evidence of the role of endogenous H_2_S in the maintenance of vascular health. Human placental CSE and CBS expression and H_2_S production are reduced in PE [338,417]. The inhibition of placenta CSE/H_2_S production by PAG results in PE-like symptoms in mice due to impaired placental angiogenesis [338]; H_2_S supplementation using NaSH or GYY4237 reverses sFlt-1-induced hypertension and proteinuria in rats [338,418]. Moreover, spiral artery CSE downregulation mediated by miR21 upregulation is associated with abnormal UA remodeling and umbilical artery Doppler profiles in IUGR and PE pregnancies [419]. These studies have established the causal role of the CBS/CSE-H_2_S system in uterine and placental vascular development and function in pregnancy and have also provided a premise that H_2_S is a potential therapeutic target for PE and IUGR.

### 5.2. Therapeutic Considerations

The clinical manifestations (i.e., new onset hypertension and proteinuria) of PE are diagnosed after the 20th week of gestation. Current treatments for PE are aimed at normalizing the blood pressure rather than targeting the placenta pathology itself, but none of these are satisfactory; hypertension is reduced transiently, allowing Caesarean delivery to be set up before term. Once clinical manifestations of PE have developed, theoretically, it should be too late to correct placenta defects as a target for the therapeutic development for PE, since the formation of a fully functional placenta has been completed before the 20th week of gestation. Improving UtBF provides an attractive alternative for managing clinical PE (i.e., hypertension) to extend gestation safely so that premature delivery can be avoided.

Short-term E_2_β treatment in PE women reduces clinical symptoms in association with decreased oxidative stress by reducing the production of oxidants, i.e., O^−^, H_2_O_2_, NO_2_^−^, and ONOO^−^ [420,421]. However, too many off-target effects prevent estrogens being used as a therapeutic target for PE. Animals treated with l-NAME to inhibit endogenous NO production develop PE-like symptoms [422]. Likewise, the NO pathway is perhaps the most studied as a potential therapy target for PE, including endogenous *S*-nitrosothiol GSNO, the NO precursor l-arginine, the PDE5 inhibitor sildenafil, etc. [423]. GSNO improves endothelial function, reduces platelet aggregation, and promotes UtBF [423,424,425]. In a randomized controlled trial, it was shown that pregnant women at high risk of PE benefit from dietary supplementation with a combination of l-arginine and antioxidants, but not l-arginine or antioxidants alone [426]. Subsequently, the beneficial effect of l-arginine has been confirmed in seven controlled randomized trials showing that l-arginine supplementation is manifested with a significant reduction in the risk to pregnant women with established or suspected PE [427]. Bolstering endogenous NO signing using sildenafil and its derivatives has been investigated extensively for treating PE. Despite its promising results in many preclinical animal studies, human studies have shown a 4-day extension in gestation in severe PE patients [428]. However, a very recent clinical trial showed the beneficial effects of sildenafil in treating pregnant women with a high risk of IUGR; however, unfortunately, the trial was halted because infants born to women receiving the drug died from pulmonary hypertension [429].

Statins are medicines that were originally developed for hypercholesterolemia [430], with a known protective effect on vascular EC [431,432] and a stimulatory effect on CO-synthesizing enzyme HO expression [433]. Mice treated with statins have shown to possess elevated levels of CO production and increased levels of plasma antioxidants, which are abrogated by HO inhibitors, indicating that the HO/CO pathway may mediate the vasodilatory actions of statins [434]. Statins do not seem to be teratogenic; however, the available evidence only shows limited, yet promising, efficacy in treating and preventing PE [435].

Wang and colleagues have provided preclinical evidence for targeting the placental CSE/H_2_S pathway as a promising approach for treating PE [338]. They found that the inhibition of the CSE/H_2_S pathway by PAG induces hypertension and liver damage, promotes abnormal labyrinth vascularization in the placenta, and decreases fetal growth in mice; GYY4137 supplementation inhibits circulating soluble Flt-1 levels and restores fetal growth in PAG-treated dams [338]. Another study provides further preclinical evidence of the therapeutic potential of H_2_S in treating PE, because 8 days of treatment with NaHS (50 µmol/kg) decreased hypertension, proteinuria, and endotheliosis in sFlt-1-induced PE rats, in association with increased kidney VEGF production [418]. However, to date, no clinical trials have been reported on targeting H_2_S for PE. Notably, the consumption of natural products such as garlic is associated with a lower incidence of hypertension due to its antioxidant properties and inhibition of platelet aggregation [436]. Garlic-rich diets contain high organic polysulfides that exert H_2_S effects, which have been proposed to prevent and ameliorate cardiovascular disease [437]. Currently, clinical trials are being undertaken to test the beneficial effects of these kinds of “medical foods” or nutraceuticals, such as the slow-releasing H_2_S donor diallyl trisulfide (DATS, [366]) and sulfur-releasing sodium polysulthionate (SG-1002, [332]) in cardiovascular diseases. It would be interesting to test whether these natural “H_2_S” donors have beneficial effects on PE.

## 6. Conclusions

Estrogens are pluripotent steroid hormones that regulate numerous physiological processes in the human body. Estrogen signaling in the uterine vasculature remains a fascinating yet understudied research area. The biological effects of estrogen are mediated by multiple ERs that activate multiple signaling pathways under the influence of physiological (e.g., pregnancy) and pathophysiological (e.g., preeclampsia) conditions, adding more complexity to the understanding of estrogen-induced uterine vasodilation. Estrogens and their metabolites are potent uterine vasodilators in the nonpregnant state, and these effects are further augmented during pregnancy. The vasodilatory effects of estrogens in the uterine and selected systemic arteries are well established as being largely mediated by ER-mediated genomic and nongenomic pathways, they involve the enhanced production of NO by the EC and CO and H_2_S by both EC and SMC locally, in specific vascular beds. However, the mechanisms of how each of the NO, CO, and H_2_S systems is regulated and how they interact to participate in estrogen-induced uterine vasodilation in pregnancy are yet to be discovered. Nonetheless, enhanced local UA production of gasotransmitters provides the promise that targeting these signals can be used to develop therapeutics for preeclampsia and IUGR. However, to make this promise into a reality, more research in this area is needed, because targeting these pathways has hitherto failed or only achieved minimal success in treating or preventing PE and IUGR. Nonetheless, the H_2_S pathway in pregnancy is the least studied among the three. The emerging role of H_2_S in estrogen-induced uterine vasodilation in pregnancy should be studied further so that the physiological role of enhanced endogenous H_2_S and the pathophysiological role of dysregulated H_2_S in pregnancy complications such as PE and IUGR can be established. Furthermore, the interactions among NO, CO, and H_2_S should be taken into special consideration in future studies aiming to develop therapies for pregnancy complications.

## Figures and Tables

**Figure 1 ijms-21-04349-f001:**
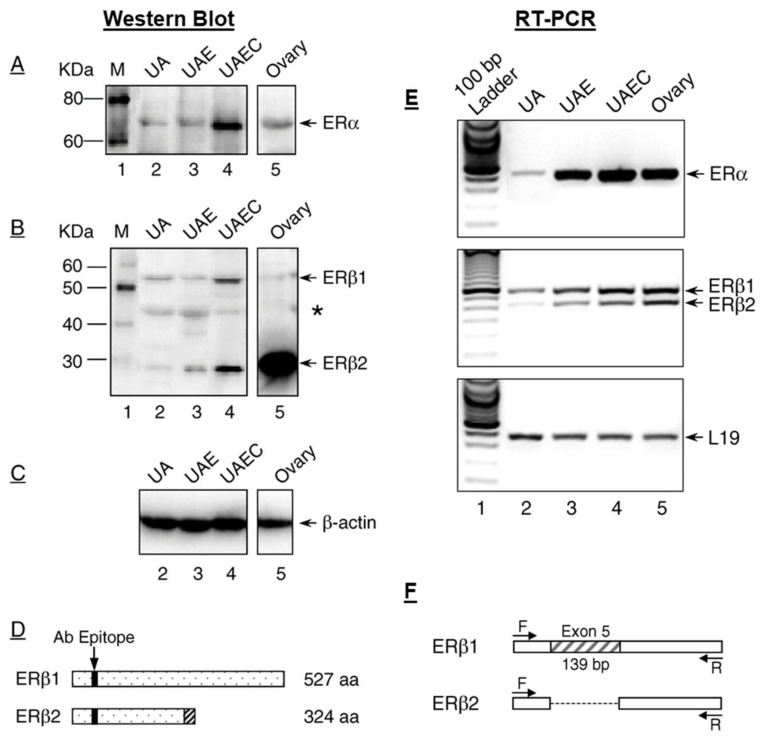
Uterine artery estrogen receptors. Estrogen receptor (ER)α (**A**) and ERβ (**B**) proteins in the protein extracts of intact uterine artery (UA), purified UA endothelium (UAE), cultured UA endothelial cells (UAEC), and ovary from pregnant ewes detected by Immunoblotting with epitope specific antibodies. β-actin (**C**) was used as loading control. (**D**) A diagram representing the truncated form of ERβ2 that results from the splicing deletion of exon 5. The shadowed box represents the amino acid sequences encoded by different reading frame. (**E**) ERα and ERβ mRNAs detected by RT-PCR with the ribosomal protein L19 as loading control. (**F**) A diagram shows a 139bp deletion of Exon 5 in ERβ2 mRNA vs. the native ERβ1. Uterine artery (UA); UA endothelium (UAE); UA endothelial cells (UAEC); amino acid (aa). Bands marked with * may indicate additional truncated forms of ERβ. Adopted from Liao et al. [71].

**Figure 2 ijms-21-04349-f002:**
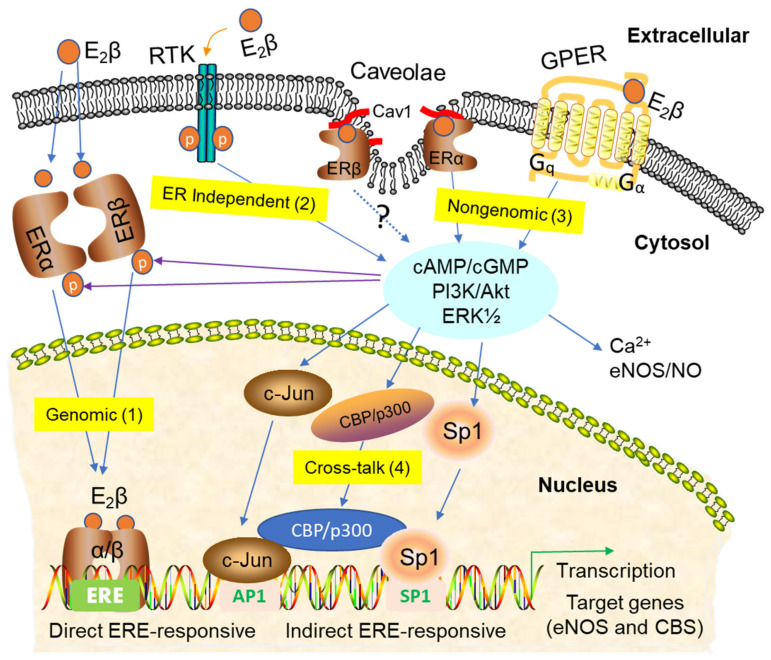
Signaling pathways of estrogen actions. There are different signaling mechanisms that mediate estrogen actions, including (1) nuclear actions: estrogen receptor (ERs) bind estrogens; liganded ERs translocate to the nucleus where they form homo- or heterodimer transcription factors to induce the transcription of estrogen-responsive genes with or without estrogen-responsive elements (EREs), (2) ER-independent estrogen signaling: estrogen transactivates receptor tyrosine kinases (RTKs, i.e., epidermal growth factor receptor and fibroblast growth factor (EGFR and FGFR)) to initiate ER-independent signaling such as an increase in second messengers cAMP and guanosine monophosphate (cGMP) the and activation of protein kinase cascades such as extracellular signal-regulated kinases ½ (ERK½) and phosphoinositide 3-kinases (PI3K)/protein kinase B (AKT), (3) extranuclear nuclear nongenomic actions: estrogens binds to plasma membrane ERs, including ERα and ERβ, and G-protein-coupled estrogen receptor (GPER), resulting in increase in second messengers and activation of protein kinase cascades, and (4) interactions between nuclear and extra-nuclear estrogen signaling pathways: ER-independent estrogen signaling via transactivation of RTKs and plasma membrane ER-dependent nongenomic estrogen signaling can interact with nuclear estrogen signaling via protein kinase-mediated ER phosphorylation and activation of gene transcription via ER-interacting transcription factors such as activator protein-1 (AP-1) and specificity protein-1 (SP-1).

**Figure 3 ijms-21-04349-f003:**
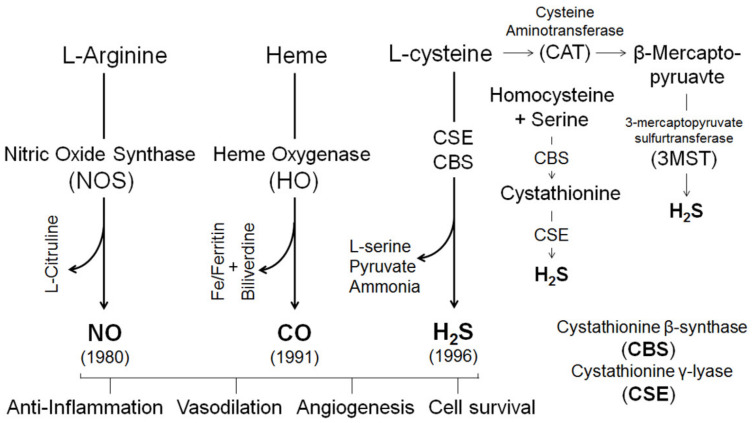
Gasotransmitters: biosynthesis and function. Nitric oxide (NO), carbon monoxide (CO), and hydrogen sulfide (H_2_S) were identified as gasotransmitters in 1980, 1991 and 1996, respectively. NO is synthesized by a family of NO synthases (eNOS, nNOS or iNOS) via converting l-arginine to l-citrulline. CO is synthesized by heme oxygenases (HO-1 or HO-2) by oxidizing heme to produce CO along with biliverdin and iron/ferritin. H_2_S is mainly synthesized by cystathionine β-synthase (CBS) or cystathionine-γ-lyase (CSE) by converting l-Cysteine to l-Serine or Pyruvate and NH_4_^+^; H_2_S can also be produced by condensation of homocysteine and serine into cystathionine by CBS and by CSE to produce H_2_S. In addition, cysteine aminotransferase (CAT) and 3-mercaptopyrivate sulfurtransferase (3MST) can also catalyze the biosynthesis of H_2_S from l-Cysteine via the intermediate product as β-Mercapto-pyruvate. The gasotransmitters NO, CO and H_2_S act to regulate cell survival, anti-inflammation, cardioprotective angiogenesis and vasodilation.

**Figure 4 ijms-21-04349-f004:**
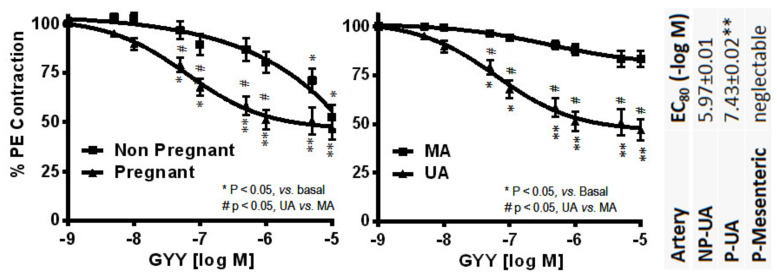
Hydrogen sulfide (H_2_S) functions as a “new” uterine artery dilator. Endothelium intact primary uterine (UA, *n* = 12–18) and secondary mesenteric (MA, *n* = 12) arteries isolated from pregnant (P) (gestation day 20) and nonpregnant (NP) Sprague–Dawley rats. Artery rings were preconstricted with 10^−6^ M phenylephrine and exposed to a slow releasing H_2_S donor morpholin-4-ium-4-methoxyphenyl(morpholino) phosphinodithioate (GYY4137) (10^−9^–10^−5^ M). * *p* < 0.05, ** *p* < 0.01, for differences within the same dose; # *p* < 0.05 vs. NP-UA, or vs. P-MA in the same dose. The potency of GYY4137 in dilating each artery is compared using EC_80_, which is calculated as the log (M) of the dose that induces 80% dilation. ** *p* < 0.01 vs. NP-UA. Adapted from Sheibani et al., Biol. Reprod. 2019; 96:664–672 (37).

**Table 1 ijms-21-04349-t001:** Uterine miRNAs whose expression is regulated by estradiol-17β (E_2_β).

miRNA	Regulation	Tissue/Cell	References
miR-451	↑	Mouse uterus	[199]
miR-429	↑	Mouse uterus	[198]
miR-155			
miR-181b	↓	Mouse uterus	[198]
miR-204			
miR-21	↓	Human endometrial glandular epithelial cells	[204]
miR-20a			
miR-21	↓	Human leiomyoma SMC	[204]
miR-26a			
miR-30b	↑	Human UVEC	[205]
miR-487a			
miR-4710			
miR-501-3p			
miR-378h	↓	Human UVEC	[205]
miR-1244			

↑, upregulation; ↓, downregulated.

**Table 2 ijms-21-04349-t002:** Uterine miRNAs, their proposed function and transcriptional targets.

miRNA	Tissue/Cell	Functions	Targets	References
miR-199a	human telomerase reverse transcriptase-immortalized human myometrial (hTERT-HM) cell line	Contractility	Cyclooxygenase-2 (COX-2)	[201]
miR-214				
miR-203	Rat uterus	Proliferation	Zinc finger and BTB domain containing 20 (Zbtb20), Alkaline Ceramidase 2 (Acer2)	[202]
	Endometrial carcinoma	Migration		
miR-451	Human endometriotic lesions	Migration	Macrophage Migration Inhibitory Factor (MIF)	[210]
miR-218-5p	Extravillous cytotrophoblast (EVT)	Trophoblast differentiation	Transforming growth factor β 2 (TGFβ2)	[211]
		Vessel remodeling		
miR-204	Human endometrial cancer-1 (HEC-1A) cell line	Migration	Forkhead box C 1 (FOXC1)	[212]
		Invasion		
miR-365	Trophoblasts	Cell cycle	Mouse double minute 2 (MDM2), p53	[213]
		Apoptosis		
miR-376c	Trophoblasts	Proliferation	TGFβ2	[214]
		Invasion		
miR-17	Placenta	Angiogenesis	Ephrin-B2	[215]
miR-20a			Ephrin type-B receptor 4 (EPHB4)	
miR-20b				
miR-210	Ovine UA	Relaxation	Tet methylcytosine dioxygenase 1 (TET1)	[203]

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
