# Peer review of "Estrogen Receptors and Estrogen-Induced Uterine Vasodilation in Pregnancy"

_ijms, 2020, doi:10.3390/ijms21124349_

Round 1

Reviewer 1 Report

Review of the manuscript untitled:

“Estrogen receptors and estrogen-induced uterine vasodilatation in pregnancy”

In this manuscript the authors presented a summary on current knowledge of estrogen/estrogen receptors signaling pathways regulating uterine vasodilatation in pregnancy.

This review is of interest and overall well conducted. However, some modifications have to be done before publication.

-First of all, the manuscript needs to be edited for english, style format (text underlined, different font etc..). As an example, different sections are entitled with 3.2. Moreover there are too many abreviations. (in the abstract to play and not to plays).

  • Lane 104, ERa is also expressed in breast.
  • Lane 105 : I do not think that ERa46 is the most studied. ERa36 should be presented, as it is much more studied than pER etc…
  •  

Lane 122 : GPER should be introduced here.

Lane 132 : in in

Figure 1 : knowing that a lot of ERa antibodies are not specific, please include a control with an siRNA

ICI should be introduced as an ER antagonist.

Please modify the sequence of the pathways in the figure legend.

Lane 349 ; any transmembrane domain.

Lane 411 : this section is about estrogen independent action, not ERa independent.

Lane 432 : What is the rationale to focus on sumoylation, what about the other PTMs ?

Lane 550 : highly reactive with reactive.

Author Response

In this manuscript the authors presented a summary on current knowledge of estrogen/estrogen receptors signaling pathways regulating uterine vasodilatation in pregnancy. This review is of interest and overall well conducted.

Thank you for these comments.

First of all, the manuscript needs to be edited for english, style format (text underlined, different font etc..). As an example, different sections are entitled with 3.2. Moreover, there are too many abreviations. (in the abstract to play and not to plays).

Yes, we have done a thorough proof reading in R1.

  • Lane 104, ERa is also expressed in breast.
  • Yes, added.
  • Lane 105 : I do not think that ERa46 is the most studied. ERa36 should be presented, as it is much more studied than pER etc…
  • Thank you for this comment. We have added some sentences on ERa36 and vascular function.
  • Lane 122: GPER should be introduced here.
  • GPER was discussed in 3.3 in detail. We modified this place a little to help the flow.
  • Lane 132: in in.
  • One “in” is deleted.
  • Figure 1: knowing that a lot of ERa antibodies are not specific, please include a control with an siRNA.
  • We concur with this comment. However, this figure was adapted from a study we published in 2005. The ERa antibody used in the study was very clean, only gave one band of 66 kD. We authenticated the antibody by several methods including antigen depletion and using recombinant ERa as a controls as reported in our other paper (Smith A, et al. Gender-Specific Protection of Estrogen against Gastric Acid-Induced Duodenal Injury: Stimulation of Duodenal Mucosal Bicarbonate Secretion. Endocrinology. 2008; 149:4554-4566).
  • ICI should be introduced as an ER antagonist.
  • We added ICI as the ER antagonist in line 86.
  • Please modify the sequence of the pathways in the figure legend.
  • Changed as needed.
  • Lane 349; any transmembrane domain.
  • Changed as needed.
  • Lane 411: this section is about estrogen independent action, not ERa independent.
  • Changed as needed.
  • Lane 432: What is the rationale to focus on sumoylation, what about the other PTMs ?
  • Yes, there are many other PTMs on ER. Nonetheless, this is not the focus of this review. Sumoylation was introduced here, just simply it alters the transcriptional activity of ER.
  • Lane 550: highly reactive with reactive.
  • Changed as needed.

Reviewer 2 Report

Minor Comments

  • Pag 3, Line 132, …..”in in”…
  • Pag 4, line 156-159 is repeated in pag 2 line 84-87, to avoid redundancy
  • Pag5, Line 204, the word “cyclic” shoud be not in bold
  • Pag9, line 384-386, revised the sentence “For instance, G1 could mimic
    the cardiac effects induced by E2
    β in rat cardiovascular system (171), and promotes the pregnancy-associated vasodilatory effects in rat UA which was blocked by G15 (102).
  • Pag 13, line 531-533, to remove the undelight
  • Pag 15, line 638, Raf-1/ERK1/½
  • Pag , line 670,    sAC-cGMP-PKG should sGC-….
  • Pag 18, line 735, to check …..” by progesterone but estrogens”.

Author Response

Reviewer 2:

Minor Comments

  • Pag 3, Line 132, …..”in in”…
  • Deleted one in as needed.
  • Pag 4, line 156-159 is repeated in pag 2 line 84-87, to avoid redundancy
  • We agree on the redundancy issue. We used this in “Introduction” just for the “flow”, but detailed description was in 156-159. We simplified line 84-87.
  • Pag5, Line 204, the word “cyclic” shoud be not in bold.
  • Changed as needed.
  • Pag9, line 384-386, revised the sentence “For instance, G1 could mimic
    the cardiac effects induced by E2β
    in rat cardiovascular system (171), and promotes the pregnancy-associated vasodilatory effects in rat UA which was blocked by G15 (102).
  • Changed as needed.
  • Pag 13, line 531-533, to remove the undelight
  • These underlight was created with PDF conversion. Changed as needed in R1.
  • Pag 15, line 638, Raf-1/ERK1/½
  • Changed as needed.
  • Pag, line 670,  sAC-cGMP-PKG should sGC-….
  • Changed as needed. Thanks.
  • Pag 18, line 735, to check …..” by progesterone but estrogens”.
  • Checked and changed as needed.